# Oxide nanolitisation-induced melt iron extraction causes viscosity jumps and enhanced explosivity in silicic magma

Francisco Cáceres [1,2] ✉, Kai-Uwe Hess [1], Michael Eitel[1], Markus Döblinger[3], Kelly N. McCartney [4], Mathieu Colombier [1], Stuart A. Gilder[1], Bettina Scheu[1], Melanie Kaliwoda[1,5] & Donald B. Dingwell [1]

Explosivity in erupting volcanoes is controlled by the degassing dynamics and the viscosity of the ascending magma in the conduit. Magma crystallisation enhances both heterogeneous bubble nucleation and increases in magma bulk viscosity. Nanolite crystallisation has been suggested to enhance such processes too, but in a noticeably higher extent. Yet the precise causes of the resultant strong viscosity increase remain unclear. Here we report experimental results for rapid nanolite crystallisation in natural silicic magma and the extent of the subsequent viscosity increase. Nanolite-free and nanolite-bearing rhyolite magmas were subjected to heat treatments, where magmas crystallised or re-crystallised oxide nanolites depending on their initial state, showing an increase of one order of magnitude as oxide nanolites formed. We thus demonstrate that oxide nanolites crystallisation increases magma bulk viscosity mainly by increasing the viscosity of its melt phase due to the chemical extraction of iron, whereas the physical effect of particle suspension is minor, almost negligible. Importantly, we further observe that this increase is sufficient for driving magma fragmentation depending on magma degassing and ascent dynamics.

Magmas ascending to the Earth's surface are subjected to decompression that drives a decrease in volatile solubility. A decrease in volatile solubility can cause exsolution and depletion of such volatiles in the melt phase, leading to an increase of the crystallisation temperature of the melt phase (*liquidus*), which in turn decreases the solubilities of the mineral phases potentially forming in such melt compositions. This shift of the *liquidus* induces nucleation and growth of crystals[1–3], which in turn changes the melt structure of the magma. The chemical changes driven by the uptake of elements during crystal nucleation and growth normally produce differentiation of the residual melt[4], which in turn drives changes in melt viscosity[5–8]. At that

stage, crystals present in a magma may also serve as sites for heterogeneous gas-bubble nucleation that facilitate further volatiles loss[9–16], as well as further increase magma bulk viscosity by loading the magma with suspended particles[17–24]. Both an enhanced bubble nucleation event with subsequent bubble growth and magma expansion, as well as an increase in magma bulk viscosity are key processes enhancing explosivity in erupting magmas[25–28].

Magmatic nanocrystals or "nanolites"[29] have been suggested to drive both heterogeneous bubble nucleation and increases in magma bulk viscosity, potentially shifting an eruption towards conditions favourable for explosive eruptions[15,30,31]. The effect of nanolite

[1]Department of Earth and Environmental Sciences, Ludwig-Maximilians-Universität (LMU) München, Theresienstr. 41, 80333 Munich, Germany. [2]Facultad de Ciencias Básicas, Universidad Católica del Maule, Avenida San Miguel, 3605 Talca, Chile. [3]Department of Chemistry, Ludwig-Maximilians-Universität (LMU) München, Butenandtstr. 5-13, 81377 Munich, Germany. [4]Department of Earth Sciences, University of Hawaiʻi at Manoa, Honolulu, HI 96822, USA. [5]Mineralogical State Collection of Munich (SNSB—Natural Science Collections of Bavaria), Theresienstrasse 41, 80333 Munich, Germany. ✉e-mail: francisco.caceres@min.uni-muenchen.de

formation on magma viscosity has raised particular attention. Viscosity measurements of magma analogues have shown a significant increase in bulk (fluid + suspended particles) viscosity even with minor loads of particles suspended in low-viscosity fluids[30]. Contrastingly, viscosity measurements in natural nanolite-bearing andesitic magma[32] have shown a less significant increase in bulk viscosity than predicted for magma analogues at comparable nanolite content. This suggests that explosive behaviour could be reached in an erupting volcano if another mechanism, such as enhanced heterogeneous bubble nucleation and resultant fast magma degassing[15], are acting together with an increase in magma bulk viscosity. Recent studies support this idea, showing that explosivity may have been driven by nanolite formation, enhancing both bubble nucleation and a viscosity increase in the 2012 submarine eruption of Havre volcano, New Zealand[33]; while nanolite crystallisation in the magma reservoir due to the intrusion of an oxidising, hot magma may have been the trigger of the 2021 submarine explosive eruption of Fukutoku-Oka-no-Ba, Japan, by enhancing bubble formation in the magma reservoir[34].

Oxide nanolites have been formed in natural products of diverse eruptive styles and magma compositions, and appear to be the most common phase crystallising at a nanometric scale in natural magmas[30,33–44]. Yet the conditions necessary to form and stabilise nanolites in magma, as well as the extent and mechanism behind the increase in magma bulk viscosity driven by nanolite crystallisation remain unclear. Here we experimentally explore the potential for oxide nanolite nucleation and growth in natural silicic magma and we measure the associated increase of magma bulk viscosity, accounting for the contributions of melt viscosity and crystals suspension.

## Results

### Experimental observations

An initial iron-rich (3.34 wt%) and nanolite-free natural rhyolite was melted at *superliquidus* temperature (1500 °C) and subjected to cooling at controlled rates of 0.1–0.5 °C min$^{-1}$ and rapid quenching in

air (>100 °C min$^{-1}$). Resultant glasses and groundmasses show none to low concentrations of oxide nanolites formed, depending on the rate of cooling applied, that are confirmed by Raman spectroscopy and magnetic analyses (Fig. 1), and consistent with previous results using the same procedure[31]. These nanolite-free (rapid quench) and low-content nanolite-bearing (slow cooling) samples (Fig. 1) were then subjected to heating at 25 °C min$^{-1}$ in a Differential Scanning Calorimeter (DSC). This permitted to simultaneously analyse the glass transition, crystallisation and/or crystal melting events occurring within the sample during heating, showing rapid and noticeable crystallisation events (exothermic peaks) immediately upon crossing the glass transition (endothermic peaks) for all the nanolite-bearing samples, whereas the initially nanolite-free sample (hereafter called rapid-quenched; Fig. 2) is lacking such event. Analyses conducted in sequence at cooling/heating rates of 25/25, 15/15 and 10/10 °C min$^{-1}$ immediately after the first heating analyses in the same DSC device, show a shift of the glass transition towards higher temperatures for the first 25/25 analyses for all the initially cooling-controlled (cc) samples (Fig. 2A–E). On the contrary, the initially rapid-quenched (rq), nanolite-free sample shows a shift of the glass transition towards a lower temperature for the first 25/25 analysis, consistent with the higher glass transition shown during the first heating due to the high cooling rate to which this sample was initially subjected during quenching. Yet all 15/15 and 10/10 analyses show consistent shifts of the glass transition towards lower temperatures for all samples compared to the first 25/25 analyses, which is normal for analyses at different cooling/heating rates[45]. A final 25/25 analysis was performed on all samples in order to compare potential shifts of the glass transition and monitor further crystallisation events between these and the first 25/25 analyses. These last analyses show insignificant shifts of the glass transition towards higher temperatures compared to the first 25/25 analyses, related to modifications in the melt composition due to crystallisation. For this last 25/25 analysis, the initially rapid-quenched sample shows a first small exothermic peak of

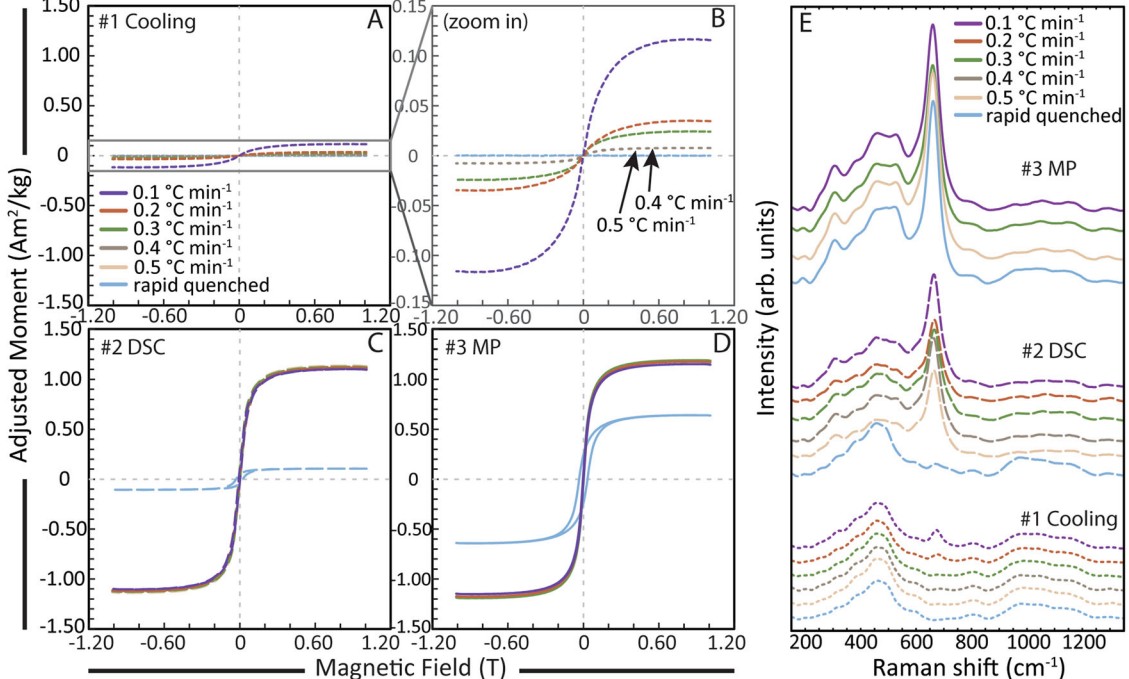

**Fig. 1 | Magnetic hysteresis loops and Raman spectra.** Hysteresis loops are normalised by sample weight and adjusted for dia-/paramagnetic contributions from: **A**, **B** controlled cooling experiments (#1 Cooling) with rates as indicated, **C** differential scanning calorimetry (#2 DSC), and **D** micro-penetration (#3 MP) viscosity analyses. **E** Raman spectra after each analysis. Note that analyses were performed in the same sequence and both magnetisation as well as the 670–690 cm$^{-1}$ Raman peak are consistently growing, indicating an increase in crystallinity. All individual normalised magnetic hysteresis curves are shown in Fig. S2. arb. units arbitrary units. Source data are provided as a Source Data file.

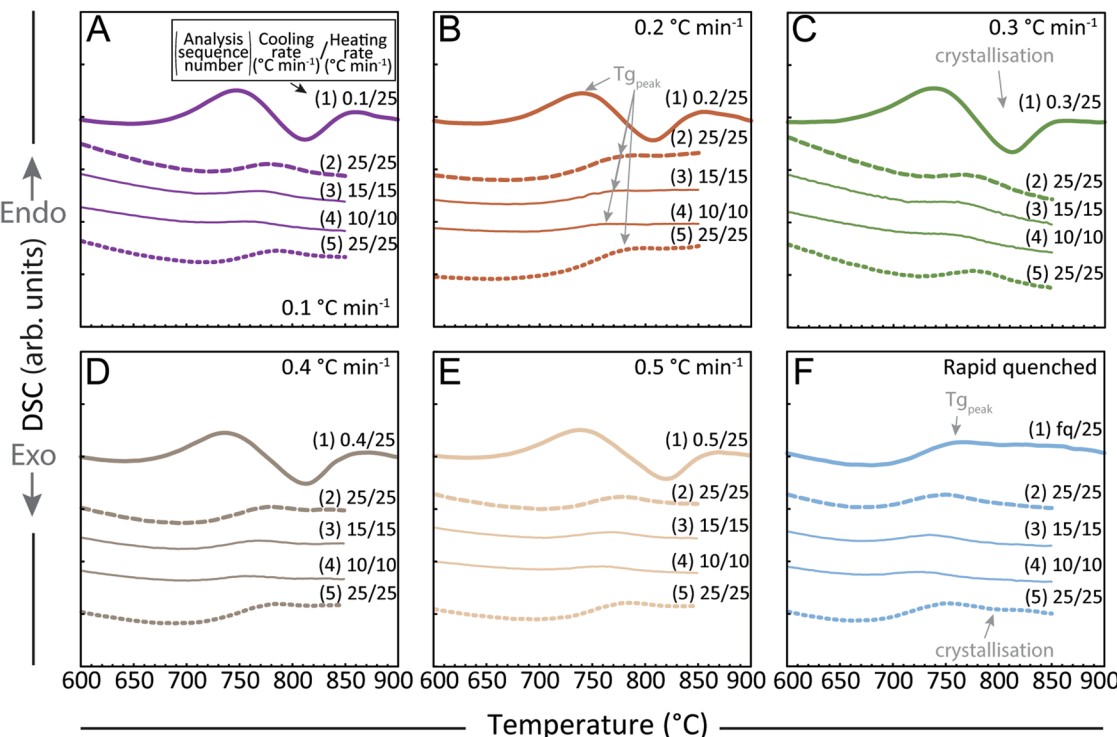

**Fig. 2 | Differential scanning calorimetry (DSC) analyses for the previously cooled (controlled cooling experiments) samples.** According to initial cooling rate, analyses correspond to the samples: **A** 0.1 °C min⁻¹, **B** 0.2 °C min⁻¹, **C** 0.3 °C min⁻¹, **D** 0.4 °C min⁻¹, **E** 0.5 °C min⁻¹, and **F** rapid-quenched (rq). Analyses were performed in the same sequence for each sample: (1) heating to 900 °C at 25 °C min⁻¹, (2) cooling at 25 °C min⁻¹ and then heating to 850 °C at 25 °C min⁻¹, (3) cooling at 15 °C min⁻¹ and then heating to 850 °C at 15 °C min⁻¹, (4) cooling at 10 °C min⁻¹ and then heating to 850 °C at 10 °C min⁻¹, (5) cooling at 25 °C min⁻¹ and then heating to 850 °C at 25 °C min⁻¹. Note that the first analyses performed with initially known slow cooling rates show a remarkably higher glass transition peaks, followed by a crystallisation (exothermic) peak. Second analyses (25/25 °C min⁻¹) show a shift of the glass transition peak to higher temperatures and insignificant crystallisation peaks, similar to those shown by the last analyses conducted at the same cooling-heating rates. This indicates that no crystallisation occurred in a significant extant in these samples after the first crystallisation event during the analyses. arb. units arbitrary units. Source data are provided as a Source Data file.

crystallisation right after the glass transition (Fig. 2F), here inferred as nanolites nucleation.

Micro-penetration (MP) viscosity measurements conducted at 875 °C for all samples after calorimetric (DSC) analyses show similar values between 9.07 – 9.16 ± 0.1 log units (viscosity in Pa s) for the cooling-controlled samples independent of the initial cooling rate. Although a slightly lower value of 8.98 ± 0.1 log units was obtained for the initially rapid-quenched sample, which is comparable to the cooling-controlled samples within uncertainty.

Magnetic hysteresis and Raman spectroscopy analyses performed after cooling (#1 Cooling), differential scanning calorimetry (#2 DSC) and micro-penetration (#3 MP) analyses allowed us to track the events and extent of crystallisation in the samples after each thermal treatment (i.e. after DSC and MP). Magnetic hysteresis loops show mainly superparamagnetic behaviour for each sample and a significant increase in the magnetic moment of the samples in the initially cooling-controlled samples after DSC analyses (Fig. 1A, B, E). This increment of the magnetic moment correlates with an increase in the characteristic Raman peak for oxide nanolites presence within the band 670–690 cm⁻¹. Yet the increase of such properties for the same samples after MP analyses is much lower compared to that produced by DSC analyses (Fig. 1C, D). On the other hand, the initially rapid-quenched sample shows a small increase in both magnetic moment and 670–690 cm⁻¹ Raman peak after DSC analyses (Fig. 1C, E), while the increase in both properties is much higher after MP analyses (Fig. 1D, E). Moreover, coercivity is much higher for the initially rapid-quenched sample, shown by the opening of the loops for the different paths at the origin, while the total magnetic moment is lower than those of the initially cooling-controlled samples, indicating both larger nanolite

crystals and lower nanolite content respectively (Fig. 1D). This last observation is also confirmed by the scanning electron images of the samples after micro-penetration analyses (Fig. 3A–C). Analyses of the Curie temperatures after MP analyses for one of the initially slow-cooled samples and the initially rapid-quenched sample show single magnetisation events at ~500 °C and ~540 °C respectively (Fig. 3E). This indicates that the oxide nanolites correspond solely to low-Ti titano-magnetite nanocrystals (Fe-Ti oxides), with slight variations of the Ti contents between both samples[46].

After viscosity analyses by micro-penetration method, the samples show crystallinities of ~1.12 vol.%, ten times higher than those observed in the samples right after the initial cooling-controlled experiments (~0.11 vol.%), but still relatively minor (Fig. 3D). The oxide nanolites do not appear as aggregates (Fig. 3A–C), which suggests that the number density of crystals remained constant as measured within ~10¹² mm⁻³ between the cooling-controlled experiments and the final state of the samples after the viscosity measurements. The only exception is the rapid-quenched sample that shows an increase from none (nanolite-free) to a volume fraction of ~0.61 vol.% and a crystal number density of ~10⁸ mm⁻³. The measured size ranges for the cooling-controlled and rapid-quenched samples after micro-penetration analyses are 5–13 nm and 52–128 nm respectively (Table S1).

**Oxide nanolites crystallisation and its effect on magma viscosity.**
The main crystallisation event of the cooling-controlled samples occurred during fast heating at 25 °C min⁻¹, evidenced by the largest exothermic peaks found in all first heating curves (Fig. 2) and the significant increases in both the magnetic moment and Raman peaks at

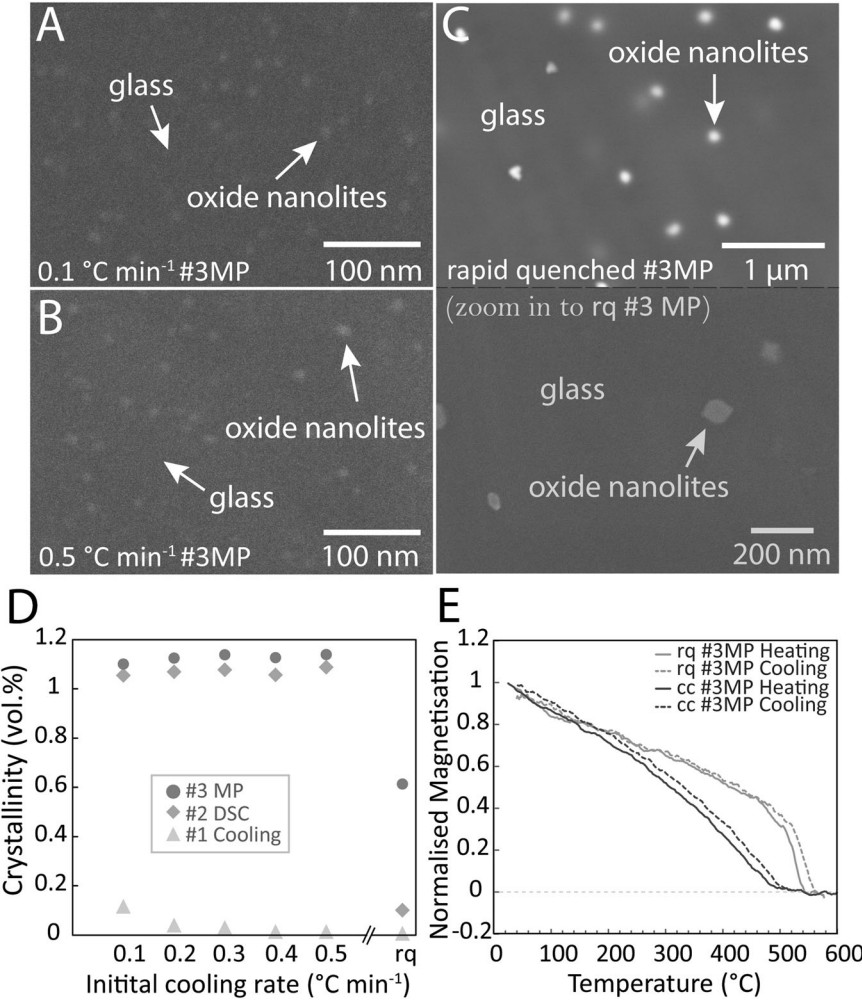

**Fig. 3 | Imaging, quantification and determination of oxide nanolites. A–C** Field emission-scanning electron microscope (FE-SEM) images of the nanolite-bearing samples. Shown are the samples after micro-penetration analyses for the initially slowly cooled samples (cc) at rates of 0.1 and 0.5 °C min⁻¹, as well as the sample subjected to rapid quenching (rq) and that was originally nanolite-free. For this sample (rq), nanolites nucleated during heating of the last calorimetry analysis (DSC) and grew during micro-penetration analysis, evidenced by both magnetic and Raman analyses (Fig. 1). All images acquired after micro-penetration (MP) analyses. **D** Crystallinities at each step of analyses, determined by magnetic analyses and calibrated by crystallinities determined from SEM images (see "Methods"). Crystallinity data can be found in Table S1. **E** Curie temperatures of the samples above with 0.61 vol.% (rq) and 1.12 vol.% (cc) crystals. Curie temperatures were determined after micro-penetration analyses, showing no change between heating and cooling curves, indicating no major change in the stoichiometry of the Fe-Ti oxides. Curie temperatures confirm that nanocrystals present correspond to titanomagnetite crystals, with slightly higher titanium content for the crystals in the samples with 1.12 vol.% (cooling-controlled) compared to that with 0.61 vol.% (rapid-quenched). Source data are provided as a Source Data file.

the 670–690 cm⁻¹ band (Fig. 1). This crystallisation capacity during heating is much higher than during slow cooling of a magma, shown by both the speed and the extent of nanolite growth. During slow cooling of rhyolite magma, nanolites formation (nucleation and growth) in a low extent requires timescales of ~$10^4$ s[31]. Here we observe that fast heating is effective for growth of nucleated crystals (for cc samples), occurring in timescales as low as ≥$10^0$ s, but ineffective for crystal nucleation, evidenced by the originally crystal-free melt (rq sample) that nucleated slightly only during the fifth heating cycle of DSC analyses (Fig. 2F). On the other hand, slow cooling shows to be effective for nucleation of crystals, but not for their growth in silicic magma. Further, nanolite crystallinities reached during heating are approximately one order of magnitude higher than during cooling (Fig. 3D). Magmas that are subjected to slow cooling such as lava domes and shallow magma plugs can be subjected to heating by friction on the conduit walls, viscous heating or magma inputs from deeper in the conduit[47], while magmas slowly ascending in the conduit are also subjected to slow cooling[31]. As shown during the heat treatments of the samples in this study, both processes are, indeed, capable of inducing

oxides nanolites formation in iron-bearing silicic magma in the time-scales fore mentioned.

Melt viscosities of the nanolite-free (rq) and nanolite-bearing (cc) samples were determined from calorimetric analyses[45] considering the glass transition peaks at different rates and a rectified shift factor (see "Methods"). These viscosities were compared to those bulk viscosities (melt + oxide nanolites) measured by micro-penetration method (Fig. 4A). Melt viscosities obtained were fit by the non-Arrhenian Vogel-Fulcher-Tammann (VFT) equation, for the initially cooling-controlled sample with highest nanolite content (0.1 °C min⁻¹) and separately for the rapid-quenched sample. These fits of melt viscosity can then be extrapolated to higher temperatures in order to obtain melt viscosity at comparable conditions to those measured by micro-penetration. The extrapolation of the melt viscosity for the nanolite-bearing sample predicts a melt viscosity at 875 °C of $10^{8.99}$ Pa s, while micro-penetration shows an average bulk viscosity of $10^{9.12±0.1}$ Pa s (Fig. 4), i.e. a maximum difference of only ~0.13 log units that accounts for the effect of suspended crystals on the bulk viscosity, neglecting the insignificant increase in crystal content between DSC and MP analyses

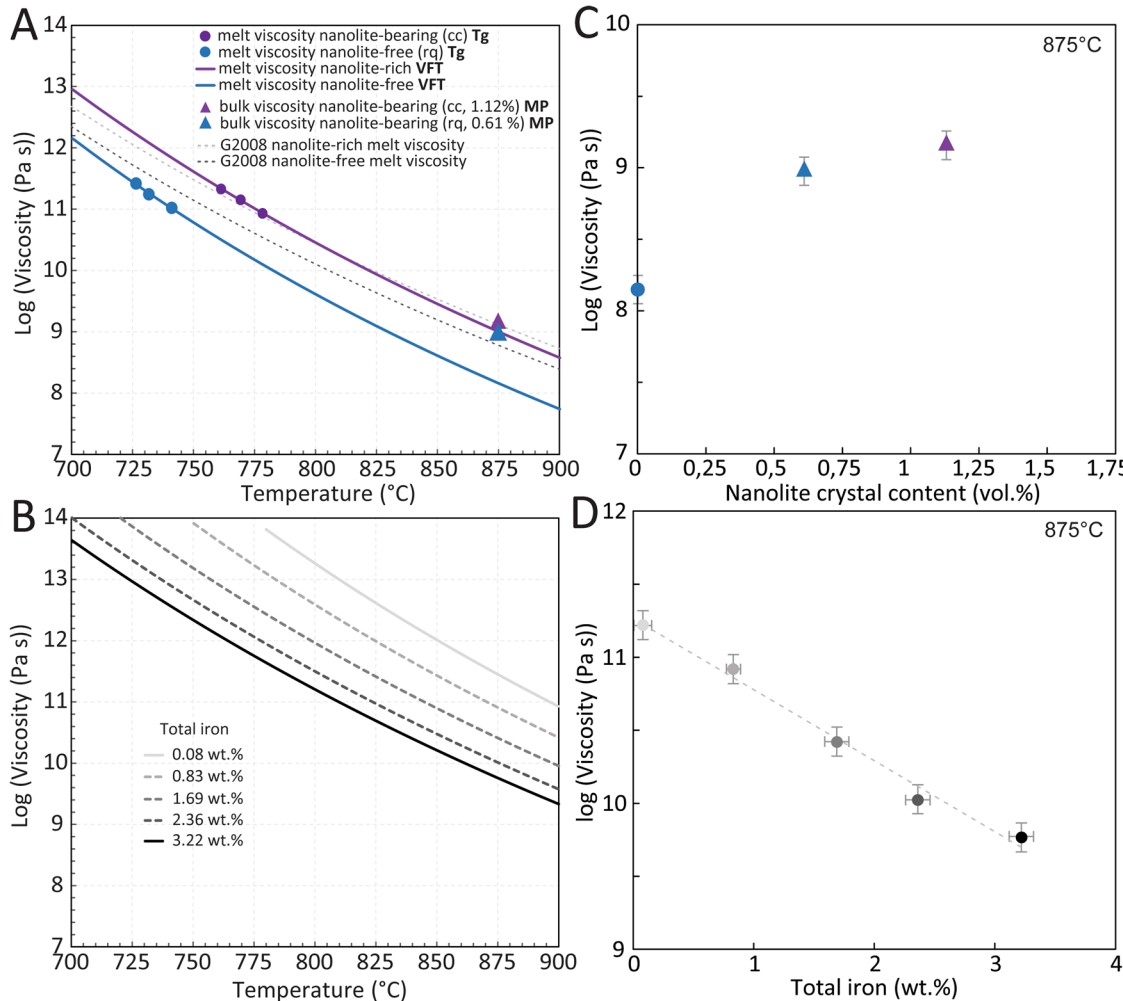

**Fig. 4 | Viscosities of nanocrystal-free melt and nanocrystal-bearing magma.**
**A** The viscosity of the nanolite-free melt (blue), the melt viscosity of the nanolite-bearing magma (purple) and the bulk magma viscosities measured with micro-penetration for the nanolite-bearing magmas (triangles): 0.61 vol.% nanolites in blue and 1.12 vol.% nanolites in purple. Dotted lines show the melt viscosities calculated by Giordano et al.[6] for nanolite-free and nanolite-bearing melts. Nanolite-bearing melt was calculated from a simulation in rhyolite-MELTS[64] when reaching a 1.12 vol.% nanocrystals. Note the shifts between the viscosity of the nanolite-free melt and the modelled viscosity, as well as the concordance between the modelled viscosity and the measure bulk viscosity of the nanolite-bearing magma. The ±0.1 log-units uncertainty associated to the viscosity measurements is shown within the symbols size (see "Methods" section for details). **B** Non-Arrhenian fits for the viscosities of synthesised rhyolite melts with variable iron concentrations. Total iron as $Fe_2O_3$ according to normalised values in Table 1. Variable iron concentrations represent approximate percentages with respect to the original natural rhyolite composition (3.34 wt%): 0, 25, 50, 75 and 100%. Complete curves including high-temperature and low-temperature viscosity data points can be found in Supplementary Material. **C** Bulk magma viscosity vs. nanocrystal content at 875 °C. Blue dot represents the extrapolated viscosity obtained from the DSC-derived viscosity. The ±0.1 log-units uncertainty associated to the viscosity measurements is shown by the error bars. **D** Melt viscosity vs. iron content. Note the exponential increase in magma viscosity as the melt iron content decreases. Dotted line represents the best fitting line of the data. Uncertainty of the total iron contents represents the standard deviation of the chemical (EPMA) analyses shown in Table 1. Source data are provided as a Source Data file.

(Fig. 3D). When compared, instead, the measured bulk viscosity and the melt viscosity predicted by the fittings of the nanolite-free (rapid-quenched) sample at 875 °C ($10^{8.15}$ Pa s; Fig. 4), it becomes evident that the effect of oxide nanolite crystallisation on melt viscosity is high, of approximately one order of magnitude (~0.83 log units) with only 1.12 vol.% of crystals. Here we show that the effect of nanoparticles (nanocrystals) suspension may be negligible in natural silicic magmas, where low amounts of nanocrystals tend to form, such as those in this study (~1.12 vol.%), and that the increase of magma bulk viscosity when oxide nanolites form is driven mainly by the increase of melt viscosity due to chemical changes of the melt phase, particularly iron extraction. In this study, these effects account for a ratio of ~1:6 between the physical effect of suspended crystals and the increase of melt viscosity, respectively.

A suggested mechanism by which nanolites can increase magma viscosity is by forming aggregates, which increases the effective volume of solids in suspension[30,32,48]. Aggregates have been observed in basaltic to intermediate magmas and low-viscosity fluids, where crystals displacement in the melt, for instance, is easier compared to silicic magma. Additionally, aggregation of nanocrystals can be considered a mechanism for crystal growth in geologic environments that it is still not fully understood[49,50]. A better understanding of the forming potential of nanolites aggregates in natural magma and their effects on magma viscosity is needed. We suggest that aggregation is likely not relevant in melts from intermediate to highly acidic compositions (intermediate to high viscosities), and indeed not extant in the latter, consistent with the fact that aggregates were not observed in our experiments.

**Effect of melt iron uptake on the viscosity and explosivity of magmas.** In order to demonstrate that the viscosity increase due to nanolite crystallisation is mainly driven by iron extraction from the melt phase and secondly (and minorly) due to particle suspension, we conducted viscosity measurements of synthesised melts as proxies for the natural rhyolite melt used in the crystallisation experiments. We used variable iron contents representing approximately a 100, 75, 50, 25 and 0% of the total iron content of the natural material (3.34 wt%; Table 1). Viscosities were measured both at high-temperature (*superliquidus*) conditions using the concentric cylinder method and at low-temperature (*subliquidus*) conditions using the micro-penetration method near Tg. Non-Arrhenian VFT fitting curves between high and low-temperature analyses of these data show that the increase in melt viscosity due to a decrease in its iron content of up to ~2.5 wt% (~75% extraction) can account by itself for most of the increase in magma bulk viscosity observed while oxide nanolites

form (Fig. 4B). This mechanism was previously suggested comparing different synthesised silicic magmas with different very-high iron contents[51], yet the actual contributions from the melt chemical changes and particles suspension were not distinguished. Here we demonstrate for the first time that iron extraction from the melt is the mechanism driving the main increase in melt viscosity, and subsequently in bulk viscosity, by conducting viscosity measurements in the same synthesised magma with a systematic variation in the iron content within concentrations relevant for natural silicic magmas. Additionally, we can observe that this process of iron extraction consequently produces a more differentiated melt due to a relative increase in the total silica content (Table 1), which is in agreement with findings of a more polymerised melt after oxide nanolites form in the same natural composition[31], and the concomitant findings of structural changes when iron ($Fe^{2+}$ and $Fe^{3+}$) is subtracted from a basaltic melt[52].

**Table 1 | Chemical compositions of the studied samples**

| Oxides (wt%) | Initial | SD | Synthesised with iron content variable respect to natural rhyolite | | | | | | | | | | SD | Nanocrystal-bearing | | |
|---|---|---|---|---|---|---|---|---|---|---|---|---|---|---|---|---|
| | Natural rhyolite[a] | | Kb-Fe-0 | SD | Kb-Fe-25 | SD | Kb-Fe-50 | SD | Kb-Fe-75 | SD | Kb-Fe-100 | | | Bulk[b] | SD | Melt[c] |
| SiO$_2$ | 75.55 | 0.48 | 77.88 | 0.54 | 77.52 | 0.15 | 76.59 | 0.38 | 75.85 | 0.50 | 74.93 | 0.57 | | 75.72 | 0.49 | 77.27 |
| TiO$_2$ | 0.26 | 0.05 | 0.29 | 0.04 | 0.26 | 0.05 | 0.28 | 0.05 | 0.27 | 0.05 | 0.27 | 0.05 | | 0.26 | 0.05 | 0.13 |
| Al$_2$O$_3$ | 12.14 | 0.20 | 12.94 | 0.20 | 12.85 | 0.19 | 12.51 | 0.18 | 12.70 | 0.19 | 12.40 | 0.17 | | 12.32 | 0.13 | 12.34 |
| Fe$_2$O$_3$Tot | 3.35 | 0.17 | 0.08 | 0.03 | 0.83 | 0.05 | 1.68 | 0.11 | 2.35 | 0.12 | 3.19 | 0.11 | | 3.34 | 0.14 | 1.03 |
| MnO | 0.10 | 0.04 | 0.07 | 0.05 | 0.08 | 0.05 | 0.07 | 0.05 | 0.07 | 0.05 | 0.07 | 0.05 | | 0.08 | 0.04 | 0.10 |
| MgO | 0.11 | 0.02 | 0.13 | 0.02 | 0.12 | 0.03 | 0.11 | 0.03 | 0.13 | 0.02 | 0.11 | 0.03 | | 0.09 | 0.03 | 0.09 |
| CaO | 1.73 | 0.10 | 1.50 | 0.08 | 1.50 | 0.07 | 1.47 | 0.09 | 1.46 | 0.08 | 1.43 | 0.09 | | 1.79 | 0.09 | 1.77 |
| Na$_2$O | 4.29 | 0.09 | 4.32 | 0.09 | 4.15 | 0.15 | 4.07 | 0.14 | 4.04 | 0.17 | 3.96 | 0.23 | | 4.25 | 0.17 | 4.39 |
| K$_2$O | 2.73 | 0.09 | 2.97 | 0.05 | 2.92 | 0.07 | 2.84 | 0.08 | 2.89 | 0.11 | 2.89 | 0.09 | | 2.82 | 0.10 | 2.79 |
| P$_2$O$_5$ | 0.01 | 0.01 | 0.01 | 0.01 | 0.01 | 0.02 | 0.01 | 0.01 | 0.01 | 0.02 | 0.01 | 0.02 | | 0.02 | 0.02 | 0.01 |
| Cr$_2$O$_3$ | n.d. | – | 0.02 | 0.02 | 0.01 | 0.02 | 0.02 | 0.03 | 0.01 | 0.01 | 0.02 | 0.03 | | 0.02 | 0.03 | – |
| SO$_3$ | n.d. | – | 0.02 | 0.02 | 0.02 | 0.02 | 0.02 | 0.03 | 0.02 | 0.02 | 0.01 | 0.02 | | 0.03 | 0.02 | – |
| Cl | n.d. | – | 0.01 | 0.01 | 0.01 | 0.01 | 0.01 | 0.01 | 0.01 | 0.01 | 0.01 | 0.01 | | 0.05 | 0.02 | – |
| Total | 100.28 | – | 100.24 | – | 100.27 | – | 99.69 | – | 99.82 | – | 99.31 | – | | 100.78 | – | 99.93 |
| FeO[d] | n.d. | – | b.d. | – | 0.29 | – | 0.57 | – | 0.86 | – | 1.14 | – | | – | – | 0.46 |
| Fe$_2$O$_3$[e] | n.d. | – | n.d. | – | 0.50 | – | 1.05 | – | 1.39 | – | 1.93 | – | | n.d. | – | n.d. |
| *n* = | 10 | – | 11 | – | 15 | – | 22 | – | 22 | – | 22 | – | | 19 | – | – |
| Normalised values | | | | | | | | | | | | | | | | |
| SiO$_2$ | 75.34 | – | 77.69 | – | 77.31 | – | 76.83 | – | 75.98 | – | 75.45 | – | | 75.13 | – | 77.32 |
| TiO$_2$ | 0.26 | – | 0.29 | – | 0.26 | – | 0.28 | – | 0.28 | – | 0.28 | – | | 0.26 | – | 0.13 |
| Al$_2$O$_3$ | 12.11 | – | 12.91 | – | 12.81 | – | 12.55 | – | 12.72 | – | 12.48 | – | | 12.23 | – | 12.34 |
| Fe$_2$O$_3$T | 3.34 | – | 0.08 | – | 0.83 | – | 1.69 | – | 2.36 | – | 3.22 | – | | 3.31 | – | 1.03 |
| MnO | 0.1 | – | 0.07 | – | 0.08 | – | 0.07 | – | 0.07 | – | 0.07 | – | | 0.08 | – | 0.10 |
| MgO | 0.11 | – | 0.13 | – | 0.12 | – | 0.11 | – | 0.13 | – | 0.11 | – | | 0.08 | – | 0.09 |
| CaO | 1.73 | – | 1.50 | – | 1.49 | – | 1.48 | – | 1.46 | – | 1.44 | – | | 1.77 | – | 1.78 |
| Na$_2$O | 4.28 | – | 4.31 | – | 4.14 | – | 4.08 | – | 4.05 | – | 3.99 | – | | 4.22 | – | 4.39 |
| K$_2$O | 2.72 | – | 2.96 | – | 2.91 | – | 2.85 | – | 2.90 | – | 2.91 | – | | 2.80 | – | 2.79 |
| P$_2$O$_5$ | 0.01 | – | 0.01 | – | 0.01 | – | 0.01 | – | 0.01 | – | 0.01 | – | | 0.02 | – | 0.01 |
| Cr$_2$O$_3$ | – | – | 0.02 | – | 0.01 | – | 0.02 | – | 0.01 | – | 0.02 | – | | 0.02 | – | – |
| SO$_3$ | – | – | 0.02 | – | 0.02 | – | 0.02 | – | 0.02 | – | 0.01 | – | | 0.03 | – | – |
| Cl | – | – | 0.01 | – | 0.01 | – | 0.01 | – | 0.01 | – | 0.01 | – | | 0.05 | – | – |
| Total | 100 | – | 100 | – | 100 | – | 100 | – | 100 | – | 100 | – | | 100 | – | 100 |

Data acquired by electron probe micron-analysis (EPMA) and corresponds to an average of "n" analyses.
*SD* standard deviation, *b.d.* below detection limit, *n.d.* not determined.
[a]Data from Cáceres et al.[15].
[b]Melt + nanocrystals by EPMA in cooling-controlled (cc) sample.
[c]Melt composition from 1.12 vol.% oxide crystals in rhyolite-MELTS simulation (see "Methods").
[d]Karl-Fischer titration analyses on the post-viscosity measurements.
[e]Fe$_2$O$_3$ = Fe$_2$O$_3$Tot − 1.11*FeO.

Consequently, it can be inferred that the increase in viscosity due to oxide nanolites formation is limited by the total iron content that is extracted and initially available in the melt from where nanolites crystallise. Here we show that the initially rapid increase of magma viscosity with an increasing nanolite content diminishes considerably at higher nanolite contents, tending to a maximum threshold value (Fig. 4C). We suggest that this threshold is different for each magma composition and dependent on the amount of total iron that can be extracted. A recent study[53] showed that, in two basalts with differences in iron and titanium contents, the one with higher contents of both cations was more prone to oxide nanolite formation than the one with lower concentration of these elements. Oxide nanolite crystallisation process that noticeably increased the bulk viscosity of such basaltic magma by up to two orders of magnitude. Even though they did not show the maximum increase in viscosity that such basaltic magma may reach, their results combined with those of this study demonstrate the tendency of natural magmas over a very broad compositional range to behave similarly regarding the increase in viscosity when oxide nanolites form. A further study using basaltic melts[54] has found that viscosity noticeably increased by approximately one order of magnitude at very low crystal contents. Even though nanolites were not investigated in such study, formation of oxide nanolites has been suggested as a possible source of this viscosity increase.

On the other hand, predicted melt viscosities for iron-bearing silicic magma using a state-of-the-art model[6] (Fig. 4) are not only considerably higher than those measured in this study, but this model also predicts as melt viscosity values similar to the bulk viscosity of the nanolite-bearing magma. This shows that such model overestimates the viscosity of iron-bearing melts, for which real viscosities are considerably lower, and highlights the need for a revision of viscosity measurements of iron-bearing "melts" and the subsequent models for estimating their viscosities. This is supported by recent experimental results[55] showing that the viscosity of a basaltic melt may be lower than that previously measured and model-predicted by a factor of 2 to 5, by not considering the effect of oxide nanolites forming in such melts.

Ultimately, in order to show the effect that such increase in magma viscosity can have on the propensity of a magma to fragment, we can analyse their brittle/ductile transition. Brittle behaviour in magma can be reached when strain rates (deformation timescales) at which the magma is subjected are higher than the ability of a melt to accommodate such deformation, related to its relaxation timescale, which in turn is highly dependent on magma viscosity[25,26,28,56–59]. This criterion implies that ascending magmas in a conduit at conditions below the fragmentation threshold (i.e. with either lower strain rates or lower melt viscosities) may be driven to fragmentation by an event of oxide nanolite crystallisation (Fig. 5), depending on the degassing dynamics and the resultant strain rates produced by bubble growth and ascent of the magma towards the surface. This means that even an increase of viscosity of one order of magnitude, such as the one generated by oxide nanolite crystallisation in this study, is enough to induce explosivity in an erupting volcano with initially lower viscosity if the strain rates of the magma are close enough to those in the range for fragmentation, but not necessarily meeting them previous to nanolite formation. Adding the short crystallisation timescales for oxide nanolites mentioned beforehand, an ascending magma in a conduit is capable of increasing its viscosity and reaching fragmentation very rapidly, consequently enhancing an explosive eruption over an effusive one.

**Implications for natural magmatic systems.** Nanolites of diverse minerals types (e.g. plagioclase, pyroxene, biotite, Fe-Ti oxides) have been recognised to date in natural systems and forming different assemblages, with oxides being the common mineral present in a wide range of compositions from rhyolite to basalt[30,33–44]. Even though basaltic magmas are highly prone to oxide nanolite formation[53,55] due

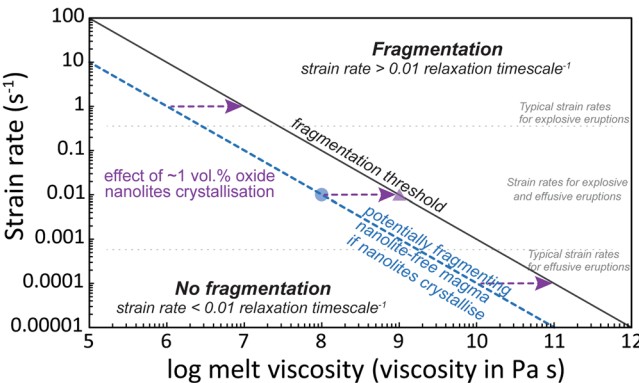

**Fig. 5 | Fragmentation criterion for strain-induced magma fragmentation.** Line shown considers a fragmentation threshold where strain rate equals to $\kappa G_\infty / \mu_m$, where $\kappa$ and $G_\infty$ are an experimental constant and the elastic modulus at infinite frequency with values of 0.01 and 10 GPa respectively[56,57], and $\mu_m$ is the melt viscosity. Considering common strain rates for silicic magma ascent[58,59], an event of oxide nanolites crystallisation can move a magma into the conditions necessary for fragmentation and explosive behaviour by increasing mainly its melt viscosity. Dotted grey line shows the strain and viscosity conditions of a magma that may potentially fragment if oxide nanolites crystallise in it, reaching the fragmentation criterion (black solid line) by increasing its viscosity by one order of magnitude. Dotted horizontal grey lines show the explosive and effusive behaviour from Gonnermann and Manga[58]. Blue circle and purple triangle show the actual viscosities of the nanolite-free melt and nanolite-bearing magma of this study plotted at the strain rates at which their jump from one viscosity to the other one would drive fragmentation and are for reference only. Such a shift in viscosity will cause fragmentation dependent on the actual strain rate and degassing dynamics of the ascending magma.

to their normally high iron contents and low viscosities, there is an increasing number of natural cases where oxide nanolites have been found in silicic magmas[33–36,41]. Most of the cases identified oxide nanolites in explosive material (i.e. pumice), and have been inferred to have influenced volcanic behaviour by enhancing explosivity. However, these studies did not quantify the net volume of oxide nanolites present. The results of this study have direct impacts into the transport properties and behaviour of silicic magmas, since it determines that for silicic systems, the effect of oxide nanolites formation is, indeed, relevant not only for enhancing heterogeneous bubble nucleation and growth[15], but also significantly increasing the already-high melt viscosity by one order of magnitude or more, even at very-low crystal fractions. This effect on viscosity, as it was shown above, can shift an ascending silicic magma into explosive behaviour, depending on its degassing dynamics and the strain rates induced by bubble growth and acceleration of the magma towards the surface. Furthermore, as it has been observed, some slowly ascending silicic magmas such as lava domes or water-depleted magma plugs can have high crystallinities[60], where in cases that their residual melts are prone to oxide nanolites formation, an enhanced explosive volcanic behaviour has also been observed[41]. Combining these observations and the results of our experimental study (Fig. 5), it can be suggested that oxide nanolites forming in the residual melt phase of highly crystalline silicic magmas may also hold the potential to promote fragmentation by increasing its magma viscosity.

We can conclude that the increase in viscosity produced by oxide nanolites formation in silicic magma is mainly driven by an increase in melt viscosity due to iron extraction from the melt, yet the contribution of non-aggregated particles suspension on the bulk magma viscosity is comparably minor. When already nucleated, oxide nanolites can grow at short timescales, rapidly increasing melt viscosity. Magmas slowly ascending in the conduit, magma plugs or magmas subjected to intrusions of other magmas are prone to oxide nanolites

formation, increasing magma viscosity and forming new sites that can potentially serve for heterogeneous gas-bubble nucleation, both effects promoting explosive behaviour. Finally, the extent of viscosity increase will depend on the extent of iron depletion in the melt due to the crystallisation of oxide nanolites. We suggest that oxide nanolites crystallisation can be a process that significantly increases melt viscosity and subsequently magma bulk viscosity in more mafic magmas due to their commonly higher iron contents compared to the ones of this study, which make such magmas more prone to oxide nanolites formation, potentially forming a higher nanolite content and extracting a higher amount of iron from the melt phase.

## Methods

### Samples preparation and cooling experiments

A natural rhyolite obsidian block from Hrafntinnuhryggur eruption at Krafla volcano in Iceland was finely powdered at <63 μm grainsize. The powder was melted at 1500 °C for 3 h in air at 1 bar atmospheric pressure in 6.8 mm-diameter and 6 mm-length Pt80-Rh20 crucibles and then rapidly quenched by removing the crucible from the furnace and placed on a room-temperature alumina plate.

Each individual Pt-Rh crucible containing remelted material was placed in an optical dilatometer at the Department of Earth and Environmental Sciences, LMU Munich, and heated again for 3 h at 1500 °C. After dwelling, each sample was cooled down to 400 °C at a highly controlled rate between 0.1–0.5 °C min⁻¹, and then the furnace of the optical dilatometer was switched off. A similar procedure was shown beforehand[31] to crystallise oxide nanolites in magma of the same chemical composition, so it was applied here. After cooling experiments, the samples were drilled out of the crucibles using a 3 mm-diameter diamond coring tool and then cut into 3 mm length cylinders using a 200 μm-thick diamond-coated wire saw, that were used for calorimetric (DSC) and viscosity (MP) analyses, as well as for Magnetic Hysteresis and Raman analyses before and after DSC and MP.

### Raman analyses

Raman spectra were obtained after cooling experiments, calorimetric analyses (DSC) and viscosity analyses (MP). Analyses were performed using a confocal HORIBA Jobin Yvon XploRa micro-Raman spectrometer from the Mineralogical State Collection Munich (SNSB). A pure silica standard was used for calibration. All spectra were acquired using a green Nd:YAG-Laser of 532 nm wavelength and a 100LWD objective lens. The spot size of the Laser was 0.9 μm. A grating of 1200 T with a confocal hole of 300 μm and a slit of 200 μm were used. A laser attenuation of 25% provided a power at the sample surface of ~2.5 mW. Analyses were acquired with an exposure time of 30 s for three times. These conditions were replicated from a previous study[31]. The back-scattered Raman radiation was collected between 100–1500 cm⁻¹ for Cooling (#1) and DSC (#2) analyses, and between 50–1500 cm⁻¹ for MP (#3) analyses, with an error of ±1.5 cm⁻¹. Spectra were baseline-corrected for images.

### Magnetic analyses

Magnetic hysteresis analyses were performed using the 3 × 3 mm cylindrical samples in a Vibrating Sample Magnetometer (VSM) at the Department of Earth and Environmental Sciences of LMU Munich by applying fields between −1.0 and 1.0 T. Analyses were performed after Cooling (#1), DSC (#2) and MP (#3), so it was possible to track the evolution of the nanolite content in the samples after each step of analyses. Crystal contents are obtained on the base of the well-known relationship between the magnetisation and mass of the magnetic particles:

$$M^S_{sample} \times m_{sample} = M^S_{tmt} \times m_{tmt} \qquad (1)$$

Where $M^S_{sample}$ is the saturation magnetisation of the sample (measured), $M^S_{tmt}$ is the saturation magnetisation of the titanomagnetite (50 Am²/kg) and $m_{sample}$ and $m_{tmt}$ are the masses of the sample and the magnetic particles respectively. Final crystallinities were obtained using a density for titanomagnetite of 5150 kg/m³ and a calculated glass density[61] according to the glass composition in Table 1.

Curie temperature analyses were performed with a variable field translation balance (VFTB), during heating and cooling for the nanolite-bearing samples after MP analyses. Initial heating was performed from 25 to 600 °C at 25 °C min⁻¹, while cooling was at the same rate from high temperature to room temperature. A second heating and cooling analysis was performed from 25 to 580 °C in order to check for any modification induced to the crystals during the first analysis, obtaining the exact same curves and finding no modifications in the sample during the process.

### Imaging and chemical analyses (SEM and EPMA)

Scanning electron microscope (SEM) images were obtained using back-scattered electron diffraction in a Hitachi SU5000 SEM at the Department of Earth and Environmental Sciences and a FEI Helios NanoLab G3 UC Field-Emission SEM at the Department of Chemistry, LMU Munich, both using previously polished samples after micro-penetration analyses. Crystallinities and number densities of oxide minerals were calculated from the SEM images by using the software FOAMS[62]. The values for crystallinities were compared to those calculated from the magnetic analyses, obtaining equivalent results.

Melt and magma (nanolites + melt) compositions were obtained using a CAMECA SX100 electron probe micro-analyser (EPMA) at the Department of Earth and Environmental Sciences, LMU Munich, and are reported in Table 1. Analyses were acquired at an accelerating voltage of 15 kV and an electron beam current of 5 nA, using a 10 μm defocused beam in order to reduce volatile alkali loss. Calibrations were made using albite for Na and Si, rutile for Ti, orthoclase for K and Al, Fe₂O₃ for Fe, periclase for Mg, bustamite for Mn, wollastonite for Ca, Cr₂O₃ for Cr, apatite for P, anhydrite for S and vanadinite for Cl. Counting times were 20 s for peak measurements and two times 10 s for background measurements for Ti, Mn, Cr, S and Cl, while counts were 10 s for peak measurements and two times 10 s for background measurements for Na, Si, Al, Fe, Mg, Ca, K and P. Elements Na, Mg, Si and Al were analysed in sequence with TAP crystal; K, Ca, Ti using PET crystal; S, P and Cl using LPET crystal; and Fe, Mn, and Cr using LLIF crystal.

### Differential scanning calorimetry (DSC)

Calorimetric analyses were made in a Netzsch 404 Pegasus DSC device at the Department of Earth and Environmental Sciences, LMU Munich. The analysed samples correspond to those initially rapidly-quenched and slow-cooled, i.e. with none- to very-low nanolite content. The samples were initially heated at 25 °C min⁻¹, and cooled down and heated again at the same rate to obtain the 25/25 initial curve. Then the samples were cooled down at 15 °C min⁻¹ and heated at the same rate to obtain the 15/15 curve. This process was repeated again for obtaining a 10/10 and 25/25 curve. Analyses were performed in argon gas heating up to 900 °C for the first heating at 25 °C min⁻¹, and 850 °C for all other curves.

### Viscosity measurements

Bulk viscosities of the natural rhyolitic material (third analysis in sequence, #3) were measured by micro-penetration method[63] using a Netzsch TMA 402 F1 device at the Department of Earth and Environmental Sciences, LMU Munich, using 3 mm-diameter and 3 mm-length cylindric samples. These samples used were first obtained from the cooling experiments and after calorimetry (DSC) analyses. Viscosity measurements were performed in air at 875 °C by first heating at 25 °C until final temperature and held for at least 15 min in order to reach sample equilibration before penetration. Penetration analyses lasted

~15 min of duration. The device was first calibrated using a DGG-1 (Deutsche Glastechnische Gesellschaft) glass standard and each analysis has an error of ±0.06 log units of viscosity[63], considering viscosity in Pa s, that we here approximate to ±0.1 log units.

Superliquidus viscosities of the synthesised rhyolitic material with variable iron contents were performed in a concentric cylinder at the Department of Earth and Environmental Sciences, LMU Munich. Five melts were first synthesised from individual oxides and carbonate powders mixtures (Table 1) and melted into a Nabertherm® box furnace in air at 1 atm and 1600 °C, in an initially iron-saturated synthesis platinum crucible. Each sample was stirred in the same crucible using an iron-saturated platinum spindle at 1600 °C in order to homogenise the mixture and facilitate gas escape. These compositions mimic the original Krafla rhyolitic magma, varying its iron content from 100% of the original Krafla iron content to 0%, with steps of 25%. The melted and quenched (crystal-free) samples were then crushed and re-melted into 2.5 cm-diameter cylindrical iron-saturated platinum crucibles for the viscosity analyses. Viscosity measurements were performed in a Deltech® box furnace with a Brookfield DVIII+ viscometer (full torque range of 0–0.7187 mNm), using a cylindrical iron-saturated spindle at rotation speeds of 0.05–4 rpm (generally 0.05 to 2.5 rpm, with one data point at 4 rpm). The system was calibrated using a DGG-1 glass standard, with an error in each analysis of ±0.05 log units of viscosity[63], that here we also approximate to ±0.1 log units of viscosity (viscosity in Pa s). Measurements of viscosity were performed between 1275–1600 °C at steps of 25 °C, starting from the highest temperature and cooling 25 °C for each measurement until reaching 1275 °C. After viscosity measurements, all melt samples were heated again to 1600 °C in order to remove the spindle, removed from the furnace and quenched in air in order to obtain a crystal-free glass for micro-penetration analyses. Final micro-penetration analyses were performed near Tg for 3 mm-diameter and 3 mm-length cylinders obtained from the glasses after analyses in the concentric cylinder. Two cylinders were obtained for each glass sample after high-temperature viscosity analyses: one cylinder was used to determine Tg in the DSC device; the second cylinder of each glass sample was used for the micro-penetration analysis, where the sample was heated until the Tg range of temperature of each sample, taking care of not exceeding Tg in order to avoid crystallisation. The micro-penetration analyses were performed as for the natural samples (see above). The range of viscosities measured for these samples guarantee a crystal-free melt during each analysis, since the viscosity values obtained are the expected within the range of Tg (Fig. S1).

DSC-derived viscosities were obtained using the glass transitions (Tg) of the nanolite-free (rapid quenched) and nanolite-bearing (0.1 °C min⁻¹) samples after the first cooling-controlled experiments. A viscosity value was obtained using $Tg_{peak}$ temperatures according to Gottsmann et al.[45]. Shift factors, that hold a big intrinsic variance, were calculated according to the original method using the glass chemistry and a correction was applied according to the melt compositions. The differences between the DSC-calculated viscosities and the micro-penetration measurements for the synthesised samples with variable iron contents were averaged, finding that for those samples bearing iron the shift factor showed a systematic shift of 0.26–0.32 below the actual value, being the average of 0.29, so we applied this average value to the shift factors used for the calculations of the DSC-derived melt viscosities of the nanolite-free and nanolite-bearing samples. The uncertainty of these DSC-derived viscosity data points is then considered as the uncertainty of micro-penetration viscosity values of the synthesised melts, since they are significantly bigger (approximated to ±0.1) than the standard deviation of the shift factor (±0.02). Glass chemistry of the nanolite-bearing sample was obtained from a rhyolite-MELTS[64] simulation in Cáceres et al.[31], accounting for the melt composition when oxide crystallinity reached 1.12 vol.% as for the nanolite-bearing sample.

## Data availability

All data supporting the findings of this study are available within the article or in the Supplementary Information. Source data are provided with this paper.

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

## Acknowledgements

The authors acknowledge funding from the Deutsche Forschungsgemeinschaft (DFG—German Research Foundation) through the project number 457579444 (CA 2743/1-1) and ERC ADV Grant 2018

834255 (EAVESDROP). F.C. and M.C. thank the Faculty of Geoscience LMU mentoring program for support with equipment for experiments. We thank Adriana Gerz for assistance performing titration analyses and Dirk Müller for assistance with EPMA analyses.

## Author contributions

F.C. conceptualised and designed this research. F.C., K.-U.H. and D.B.D. performed viscosity analyses and interpretation. F.C., M.E., K.N.M. and S.A.G. performed magnetic analyses and interpretation. F.C., M.D. and M.C. performed textural analyses and interpretation. F.C. and B.S. collected the natural sample. F.C. and M.K. performed Raman analyses. All authors contributed to the discussion of the results, as well as to the writing and revision of the different versions of the manuscript.

## Funding

## Competing interests

The authors declare no competing interests.
