## [Peer Review File · Nature Communications]

REVIEWER COMMENTS

Reviewer #1 (Remarks to the Author):

Cáceres et al. use experiments to demonstrate how nanolite crystallisation can impact bulk magma viscosity using a natural rhyolitic composition and heating experiments on both nanolite-free and nanolite-bearing samples. Their results demonstrate that chemical changes in the residual melt following nanolite crystallisation, notably the extraction of iron, has a greater impact on magma viscosity than the nano-particle suspension, by producing a more differentiated, Si-rich and Fe-poor residual melt with higher viscosity. As magma viscosity has a significant control on conduit dynamics and magma fragmentation, this viscosity increase has implications for the explosivity of volcanic eruptions.

The manuscript is well written, clearly presented and provides interesting experimental results investigating the impact of nanolite crystallisation. However, a more developed discussion of the results is needed which provides a comparison with nanolites observed in natural products of volcanic eruptions and how the implications apply to the natural case, which may have a higher particle content and/or different melt composition. The effects of nanoparticle aggregation, shown to increase the effective volume of solid particles and the impact of the particle suspension on magma viscosity should also be discussed in further detail when evaluating if the physical impact of the particle suspension is a secondary process or not. Further discussion of the experimental results and application to the natural system will improve the discussion of the implications of the work and I recommend major revision.

Line 27: The magma viscosity increase and connection to fragmentation is not discussed further in detail in the manuscript. How would a 1 order of magnitude increase in viscosity due to nanolite crystallisation promote fragmentation? From what I see in Figure 5, this increase is within the regime where explosive and effusive behaviour can occur, depending on the strain rate. Without further discussion of the mechanism, I am not sure that the low nanolite content observed in the samples would promote a significant rheological change to induce fragmentation and explosivity, as referred to in the title. Further discussion of degassing and fluid dynamics within the conduit is required when discussing magma fragmentation and explosivity.

Line 58: The results of Okumura et al. (2022) also show that the viscosity increase due to nanoparticle formation is not as large as predicted using analogues, if agglomeration of nanolites does not occur. The effect of particle agglomeration on the contribution of nanocrystals to magma viscosity should be discussed further in the manuscript to evaluate whether the viscosity contribution of the nanocrystals is a secondary process. Although the samples in the experiments do not show agglomeration, this process is observed in natural samples and should be discussed in relation to the application of the experimental results to the natural case.

Line 135: How do the nanolite contents produced relate to observations in natural samples? Nanolite contents in natural samples are also estimated to be higher than 1vol.% (Mujin et al., 2017), often forming aggregates, which may also explain the contribution of the particle to the increase in melt viscosity. A comparison with natural samples is needed in this section to contextualise the experimental results and show that samples with a low nanolite content are also found.

Line 135: The experiments seem to produce a low nanolite content (~1 vol.%) which may also explain why the contribution of the particles is less in this case. This is also shown in analogue experiments, where there is a larger relative viscosity increase with higher particle contents and at higher strain rates. How applicable are the results of the experiments to natural samples where higher nanolite contents may be observed and is the chemical modification of the melt a primary process increasing magma viscosity for low particle contents?

Line 137-138: Please report a size range for the nanolites here

Line 138: As the nanolites in the experiments do not form aggregates, how would the effect of aggregation affect the relative influence of melt iron extraction and particle content on viscosity? Nanolite aggregates have been observed in natural samples and aggregation has been suggested as a process driving viscosity increase by increasing the effective volume of solid particles. Could greater clarification be provided for which cases melt iron extraction would become the dominant process influencing viscosity?

Line 177: I agree that the effect of the chemical change on melt viscosity is likely the dominant process here, as opposed to the particle suspension, also given the low nanolite content (~1 vol.%), which appear not to form aggregates. The results of Di Genova et al. (2020a) show that the relative viscosity increase also relates to the ability of nanoparticles to form aggregates and the consequent increase in effective volume. Do you think that the melt iron extraction process is also a dominant process increasing viscosity for nanolite-bearing magmas with higher nanolite contents (>1 vol.%) and/or aggregation, or is this process dominant at low particle volumes where nanolites are mostly separated?

Line 195-196: Di Genova et al. (2020b) show that iron oxide nanolite crystallisation can induce compositional and structural changes in mafic compositions, by extracting Fe²⁺ and Fe³⁺ from the melt. Please also discuss these results here.

Line 200: I agree that nanolite crystallisation will extract Fe and produce a more differentiated melt composition. However, could this not be a local effect on melt composition, potentially forming boundary layers around crystals, as opposed to the residual melt? The experiments crystallise only 1 vol% nanolites, so the effect on melt composition may be localised and restricted to the environment surrounding the nanocrystals. How representative is the differentiated melt composition presented in Table 1 of the residual melt following the crystallisation of nanolites? If the chemical effect is localised, how would this produce an order of magnitude increase in the bulk magma viscosity?

Line 205: Would the importance of melt iron extraction in increasing bulk viscosity change for more mafic magmas with higher iron content? The experiments in this study use a composition with only ~3 wt.% Fe₂O₃, whereas a more mafic composition could have over 10% total FeO. As these magmas are more Fe- and Ti-rich and prone to nanolite crystallisation, would the impact of the solid nanoparticles on the bulk viscosity still remain low? It is likely that these compositions would produce a higher nanolite content than the experiments. The results of Di Genova et al. (2020b) should be discussed here and the role of iron extraction and nanolite crystallisation for mafic compositions.

The effect on more mafic compositions is also referred to in the conclusion, it would be great to instead develop this point further here, before reaching the conclusions section, and discuss in greater detail the implications of the results for more Fe-rich, mafic compositions.

Line 242: The reference to non-aggregated particles here implies that the lower contribution of the particle suspension to bulk magma viscosity in this study could also be explained by the lack of aggregation for the nanolites. As aggregation has been shown to occur in natural nanolite-bearing samples, it would be best to clarify in which cases melt iron extraction becomes the dominant process causing the viscosity increase and the particle suspension a secondary process. Is this the case for particle suspensions where there is no aggregation?

Line 249: This sentence does not specifically refer to the process which would increase viscosity for more mafic, Fe-rich compositions more prone to nanolite crystallisation, and I would suggest clarifying if the authors mean there is a greater degree of iron extraction from the melt for these magmas and to contextualise this with reference to the particle content. This comparison could be introduced earlier in the discussion and developed further.

Lines 306: Could the results from the textural analysis be presented in a table, possibly in the supplementary information?

Line 311-312: Further detail is required for the EPMA analysis here. What standards were used for the analysis and what were the errors for each element analysed? Were the possible effects of glass devolatilisation evaluated and taken into account when acquiring the data? Please add these details to the supplementary information.

References

Di Genova, D., Zandona, A. and Deubener, J. (2020b). Unravelling the effect of nano-heterogeneity on the viscosity of silicate melts: Implications for glass manufacturing and volcanic eruptions. *Journal of Non-Crystalline Solids*, 545, 120248.

Reviewer #2 (Remarks to the Author):

Review of "Oxide nanolite-induced melt iron extraction causes viscosity jumps and enhanced explosivity in silicic magma" by Cáceres et al. for *Nature Communications*

This manuscript presents the results of crystallization experiments and viscosity measurements on a series of melts containing iron oxide nano-crystals. The goal is to determine the effect of ~1 volume percent nanocrystals on magma viscosity, and to determine if this effect is primarily due to the physical effect of crystals or the chemical effect on changing the residual melt composition.

The data are technically sound, obtained with appropriate techniques such as electron microscopy and magnetic hysteresis, differential scanning calorimetry, and viscometry. The data have been analysed and

interpreted carefully, and methods and results are both generally presented in sufficient detail, with the exception that uncertainties need to be calculated and plotted for the extrapolated melt viscosity in Figure 4C, on which much of the conclusions depend. Additional discussion of the temperature-time history of samples during the micropenetration viscosity experiments would be helpful in determining the possibility of nanolite formation during the measurements of the synthetic samples.

The conclusions generally appear to be valid, in that relatively small amounts of oxide crystallization can produce surprisingly high increases in viscosity. However, several claims are written in overly broad ways, for example “we show that the effect of particles (crystals) suspension is negligible in natural silicic magmas” is a vast overreach, and the idea of a maximum threshold value for the effect of nanocrystals is hardly justified based on one extrapolated datum with significant uncertainties.

Finally, the results are potentially significant and will be of interest to the community, because order of magnitude variations in viscosity do have important effects on magma dynamics. The role of nanoparticles has been a topic of intense interest for a few years now and this contribution sheds welcome additional light on how they affect the rheology of rhyolitic magmas, suggesting that in this case chemical effects are more important than physical effects.

Four specific topics that should be considered by the authors are listed below:

[1] Crystallization during viscosity and DSC measurements

Lines 135-142 “After viscosity analyses by micro-penetration method, the samples show crystallinities of ~1.12 vol.%, ten times higher than those observed in the samples right after the initial cooling-controlled experiments (~0.11 vol.%)” while “the rapid-quenched sample that shows an increase from none (nanolite-free) to a volume fraction ~0.61 vol%”.

This implies that the volume fraction of nanolites grew in all samples during the micro-penetration viscosity experiments. From the methods section, heating from 25 to 875°C took 34 minutes, followed by an isothermal hold of 15 minutes before measurement. How long did the viscosity measurements take, and was there any indication of an increase in viscosity during the measurement time? Furthermore, why was only one viscosity data point collected per sample – why not measure at 850, 875 and 900°C, which would better constrain the temperature-dependence of viscosity?

Lines 150-151 “during fast heating both nucleation (for rq sample) and growth (for cc samples) occur in timescales as low as ≥ 1 s.” But this is not true for the rq sample, which only shows a slight hint of crystal nucleation on its fifth time being heated above T_g (Fig 2F). So slow cooling stage is very effective for

crystal nucleation, and rapid heating is not. I agree with the authors that rapid heating is more effective for crystal growth (lines 151-153; Fig 3D).

Lines 187-192. Were the synthesized glasses checked for nanolite crystallization after the MP measurements? If the original natural rhyolite crystallized 0.6 vol% nanolites during the MP experiment, the synthetic rhyolites would likely also crystallize, but in different amounts based on their iron contents. That would affect the interpretation somewhat, although I doubt the overall conclusions would change.

[2] Viscosity models and measurements

Perhaps the single most important datum in the paper is the blue circle at 10^8 Pas on Figure 4C. Please check it is plotted in the right place. Inspection of Fig 4A suggests the blue dot should be at $10^{8.2}$ Pas (and $10^{8.15}$ is quoted on line 174), but it is plotted closer to $10^{8.0}$ Pas on Fig 4C. It should also be indicated that this point is extrapolated from data collected at a much lower temperature, using calorimetry data that was then converted to a viscosity value (the circles in Fig. 4A). The uncertainties in this extrapolated viscosity point need to be stated explicitly, and preferably given to 2 sigma, because the magnitude of the effect of microlites (i.e. the vertical difference between the circle and the triangles on Fig 4C) has the same uncertainties. This blue circle should have error bars on Figure 4C, because they are certainly larger than symbol size. Please also provide error bars for the triangles, which represent direct measurements. An uncertainty of ± 0.1 log unit is mentioned in the text on line 170 but it is not clear if this is the measurement uncertainty.

Line 349 / Figure S1. On the topic of converting DSC data to viscosity values, the shift factor is quoted as 0.29 log units but it is clearly slightly different for each of the four melts, so it would be better to give both the mean value and the range. These uncertainties then propagate into the VFT fit, and to its extrapolated value at 875°C.

[3] Over generalization of findings.

Lines 176-177 “Here we show that the effect of particles (crystals) suspension is negligible in natural silicic magmas”. There needs to be a qualifying statement here – the highest crystal contents in this study are 1.1 volume % and such a low volume fraction would be expected to have a negligible effect on bulk viscosity (using any of the models reviewed by Mader et al. 2013 JVGR, for example). To then say that the effect of crystals is negligible in natural silicic magmas (which can include crystal-rich dacites and rhyolites with over 50 volume % crystals) is a vast overreach. Something like “In our experiments, the effect of ~1 volume % particles (crystals) suspension is negligible, as expected based on decades of experimental work on suspension rheology”, with appropriate citations, would be better.

Lines 206-206 “we show that the initially rapid increase of magma viscosity with an increasing nanolite content diminishes considerably at higher nanolite contents, tending to a maximum threshold value”. This conclusion depends strongly on the uncertainties attached to the points in Fig 4C, but it also seems unlikely to be correct. In general higher crystal contents lead to higher viscosities, for both chemical and physical reasons. There is no reason why there would be a maximum threshold for the effect of crystals, whether large or small, on magma viscosity.

[4] Additional support from previous results

Lines 249-251 several experimental rheology studies have found surprisingly large viscosity increases in basaltic magma viscosity at relative low crystal fractions (observed at micro but not nano scale). See discussion in Morrison et al. 2019 Icarus, for example. Perhaps nanocrystals can explain these results, because the effect appears to be of similar magnitude (~1 order of magnitude viscosity increase). This may belong around line 210-217 instead, and would strengthen the final sentence of the manuscript's conclusion.

Minor comments:

Line 179 it is not immediately clear what the 1:6 ratio is, especially as it is written as “crystals suspension : melt”. I am not sure if the authors intend to say “the increase of magma bulk viscosity is driven mainly by the increase of melt viscosity due to chemical changes of the melt phase, particularly iron extraction, accounting for 6-fold increase in the viscosity of the suspension relative to the uncrystallized melt.” Or whether they meant “the increase of magma bulk viscosity is driven mainly by the increase of melt viscosity due to chemical changes of the melt phase, whose effect is 6 times greater than the physical effect of the rigid crystals.”

Line 277 Change “form” to “from”

Line 585 Correct spelling of “triangles”

Line 595 I suggest removing the dotted line. The reader does not need a line to guide the eye, there are three large symbols to do that already. There are other lines that could be drawn between these symbols; in this case, to choose one does not guide so much as impose an interpretation, especially because the uncertainties are not shown for the first symbol.

Lines 610-613 Scarani et al. 2022 Comms Earth Env also suggest strong rheological effects of nanolites on basaltic magma viscosity. This is appropriately discussed in the main text, around line 210-217, but it does not also belong in the caption to figure 5.

In what follows we list the reviewers' comments in grey colour and our replies in blue colour. Referential manuscript changes are given in **bold blue**.

All changes made to the manuscript file, based on the reviewers' comments and additional changes that we believe improved the quality of the manuscript, appear with track changes (underlined in red colour) within the corresponding manuscript file.

Reviewer #1:

Cáceres et al. use experiments to demonstrate how nanolite crystallisation can impact bulk magma viscosity using a natural rhyolitic composition and heating experiments on both nanolite-free and nanolite-bearing samples. Their results demonstrate that chemical changes in the residual melt following nanolite crystallisation, notably the extraction of iron, has a greater impact on magma viscosity than the nano-particle suspension, by producing a more differentiated, Si-rich and Fe-poor residual melt with higher viscosity. As magma viscosity has a significant control on conduit dynamics and magma fragmentation, this viscosity increase has implications for the explosivity of volcanic eruptions.

The manuscript is well written, clearly presented and provides interesting experimental results investigating the impact of nanolite crystallisation. However, a more developed discussion of the results is needed which provides a comparison with nanolites observed in natural products of volcanic eruptions and how the implications apply to the natural case, which may have a higher particle content and/or different melt composition. The effects of nanoparticle aggregation, shown to increase the effective volume of solid particles and the impact of the particle suspension on magma viscosity should also be discussed in further detail when evaluating if the physical impact of the particle suspension is a secondary process or not. Further discussion of the experimental results and application to the natural system will improve the discussion of the implications of the work and I recommend major revision.

Reply: We thank the reviewer for pointing out that the link between our results and natural systems, as well as the potential role of nanoparticle aggregation were not deeply discussed in our first version of the manuscript. We have taken into account their comments and we ~~have~~ included them in the new version of the manuscript, which we consider improved greatly following the reviewer's suggestions. Detailed replies to these points, are given point-by-point in what follows.

Line 27: The magma viscosity increase and connection to fragmentation is not discussed further in detail in the manuscript. How would a 1 order of magnitude increase in viscosity due to nanolite crystallization promote fragmentation? From what I see in Figure 5, this increase is within the regime where explosive and effusive behaviour can occur, depending on the strain rate. Without further discussion of the mechanism, I am not sure that the low nanolite content observed in the samples would promote a significant rheological change to

induce fragmentation and explosivity, as referred to in the title. Further discussion of degassing and fluid dynamics within the conduit is required when discussing magma fragmentation and explosivity.

Reply: We agree with the reviewer that the discussion about the effect of the 1-order-of-magnitude increase in viscosity was presented in a condensed form. The way the increase in viscosity of one order of magnitude due to nanolite crystallization produces in magma fragmentation relates to the process of fragmentation itself. Magma fragmentation is a process that intrinsically depends on magma viscosity, since it is controlled by the ability of relaxation of the melt phase within the timescales of a strain induced to the magma (e.g. (Dingwell and Webb, 1989; Webb and Dingwell, 1990; Papale, 1999; Gonnermann and Manga, 2003). Depending on the degassing dynamics of a magma, the strain rate induced in the magma due to bubble growth and acceleration to the surface can be high enough to overcome the relaxation of the melt making it to fragment, according to the criterion shown in Figure 5 based on the papers mentioned above. Since the relaxation timescale of a magma depends primarily on viscosity, an increase in viscosity produces an increase in the timescale required for the melt to relax and then for a given strain rate the magma can reach fragmentation more easily. We would like to stress that fragmentation depends on viscosity, but not only on viscosity: it also depends on strain rate, which is influenced by the degassing dynamics of the magma.

We have now expanded this discussion following the reviewer's suggestion, and it can be found in the last paragraph of section 4 and in part of the new section 5 of the manuscript. Additionally, we have added a clarification of the conditions needed for fragmentation in the caption of Figure 5 and we have slightly modified Figure 5 in order to clarify the conditions needed for fragmentation and the effect of oxide nanolite formation.

Line 58: The results of Okumura et al. (2022) also show that the viscosity increase due to nanoparticle formation is not as large as predicted using analogues, if agglomeration of nanolites does not occur. The effect of particle agglomeration on the contribution of nanocrystals to magma viscosity should be discussed further in the manuscript to evaluate whether the viscosity contribution of the nanocrystals is a secondary process. Although the samples in the experiments do not show agglomeration, this process is observed in natural samples and should be discussed in relation to the application of the experimental results to the natural case.

Reply: Nanocrystal aggregation has been, indeed, suggested a potential mechanism for increasing the effective role of particle suspension on the rheological behaviour of magmas (DiGenova et al., 2020a). This, by increasing the effective volume of each suspension unit (aggregate). However, we would like to comment on the effect of aggregates in two ways. On one hand, nanocrystals aggregation has so far been observed only in low viscosity melts in nature, such as basaltic melts, while more differentiated melts, such as andesitic (Okumura et al., 2022), have shown little or no aggregation of nanoparticles. Furthermore, no aggregation has been observed in purely silicic melts such as rhyolitic ones. This suggests that aggregation

of nanoparticles may be a compositional-dependent phenomenon in natural magmas, likely related to the different viscosities of such melts since nanoparticles need to migrate (move in the melt) in order to form an aggregate, which can be hindered by high viscosities of the liquids in which they are suspended. For this reason, we believe that aggregation is likely not relevant in melts of intermediate to particularly acidic compositions (intermediate to high viscosities), while in the latter probably not acting. On the other hand, aggregation of nanoparticles is a still understudied phenomenon for geological fluids such as magmas, it and requires further development for interpreting their effect on viscosity, particularly when the formation of nanocrystals, as well as their effect of their formation on magma properties is still not fully understood.

This rationale has been added to the manuscript as suggested by the reviewer, and it can be found at the end of section 3 as:

"A suggested mechanism by which nanolites can increase magma viscosity is by forming aggregates, which increases the effective volume of solids in suspension (Zav'yalov et al., 2018; Di Genova 2020a; Okumura et al. 2022). Aggregates have been observed in basaltic to intermediate magmas and low viscosity fluids, where crystals displacement in the melt, for instance, is easier compared to silicic magma. Additionally, aggregation of nanocrystals can be considered a mechanism for crystal growth in geologic environments that it is still not fully understood (Ivanov et al., 2014; De Yoreo, et al., 2015). A better understanding of the forming potential of nanolites aggregates in natural magma and their effects on magma viscosity is needed. We suggest that aggregation is likely not relevant in melts from intermediate to highly acidic compositions (intermediate to high viscosities), and indeed not extant in the latter, consistent with the fact that aggregates were not observed in our experiments."

Line 135: How do the nanolite contents produced relate to observations in natural samples? Nanolite contents in natural samples are also estimated to be higher than 1vol.% (Mujin et al., 2017), often forming aggregates, which may also explain the contribution of the particle to the increase in melt viscosity. A comparison with natural samples is needed in this section to contextualise the experimental results and show that samples with a low nanolite content are also found.

Reply: Nanolite contents in natural magmas have not been reported in order to establish correlations between our experiments and those found in nature, particularly not in silicic magmas that would allow any comparison. Mujin et al. (2017) report number densities comparable to those in this study with a difference of one order of magnitude ($\sim 10^{13}$ vs. $\sim 10^{12}$ mm^{-3}), yet the nanolites formed in that andesitic composition also include plagioclase and pyroxene, so no comparison can be properly done. We have now mentioned in the manuscript (paragraph in section 5) the lack of determination of nanolite contents that exist in natural magmas as a reference to the impossibility to establish comparison with natural cases.

Line 135: The experiments seem to produce a low nanolite content (~ 1 vol.%) which may also explain why the contribution of the particles is less in this case. This is also shown in analogue experiments, where there is a larger relative viscosity increase with higher particle contents

and at higher strain rates. How applicable are the results of the experiments to natural samples where higher nanolite contents may be observed and is the chemical modification of the melt a primary process increasing magma viscosity for low particle contents?

Reply: The more nanolites are formed, the more elements are extracted from the melt. In the case of oxides, iron will be extracted proportionally to the volume of crystals formed and, hence, viscosity will increase. In this work we provide a relationship between the amount of iron in a melt and its viscosity, which was obtained by systematically measuring the viscosity in the same composition with variable known iron contents. This increase of viscosity while the iron content of the melt decreases corresponds to an exponential behaviour. So, we believe that the chemical modification of the melt that produces an increase in viscosity is a rule in magmas that is not only true for rhyolitic melts, but also for less evolved melts.

In the case of analogue experiments, the load of crystals that is possible to add is controlled by the maximum packing of the particles. In the case of natural magmas leading to volcanism, the analogue would only be valid for a limited initial part of the curve at low particle contents, since the load of crystals is controlled by the chemical availability in the melt phase capable of forming any given mineral phase, which in the case of iron, this can be consumed by the formation of oxides limiting the ability of a melt to form them. This behaviour is shown in Figure 4C, where viscosity increases to a threshold value dictated by the availability of iron in the melt.

Line 137-138: Please report a size range for the nanolites here

Reply: We have now added a size range for the nanolites. We have also added this information into the Supplementary Information, textural information table.

Line 138: As the nanolites in the experiments do not form aggregates, how would the effect of aggregation affect the relative influence of melt iron extraction and particle content on viscosity? Nanolite aggregates have been observed in natural samples and aggregation has been suggested as a process driving viscosity increase by increasing the effective volume of solid particles. Could greater clarification be provided for which cases melt iron extraction would become the dominant process influencing viscosity?

Reply: This was partially answered in the reply to the comment to Line 58. However, we would like to stress the fact that the effect of aggregation on viscosity is still poorly understood in natural geological fluids such as magmas. One can suggest that, for instance, the shapes of aggregates (likely different for each aggregate in a magma) will exert a big influence on the rheology of a magma, like it is known for isolated crystals, yet this fact is still something to be tested and proved. It is not possible so far to state the relative influence that aggregates will have on the viscosity of a magma compared to the chemical changes on the melt induced by nanolite formation, particularly since a quantification of the influence of aggregates in magmas has not been made so far. Additionally, aggregation has not been observed so far in silicic magmas.

Line 177: I agree that the effect of the chemical change on melt viscosity is likely the dominant process here, as opposed to the particle suspension, also given the low nanolite content (~1

vol.%), which appear not to form aggregates. The results of Di Genova et al. (2020a) show that the relative viscosity increase also relates to the ability of nanoparticles to form aggregates and the consequent increase in effective volume. Do you think that the melt iron extraction process is also a dominant process increasing viscosity for nanolite-bearing magmas with higher nanolite contents (>1 vol.%) and/or aggregation, or is this process dominant at low particle volumes where nanolites are mostly separated?

Reply: This has already been answered in the reply to the comment to Line 135.

Line 195-196: Di Genova et al. (2020b) show that iron oxide nanolite crystallisation can induce compositional and structural changes in mafic compositions, by extracting Fe²⁺ and Fe³⁺ from the melt. Please also discuss these results here.

Reply: What is mainly discussed in Di Genova et al., 2020b is indeed the effect of total iron extraction by extracting simultaneously the two common iron species found in melts (Fe²⁺ and Fe³⁺), generating changes in the structure of the melt by changing its compositions, as also discussed in Cáceres et al., 2021. We have added this as a sentence to the paragraph including the reference suggested by the reviewer.

Line 200: I agree that nanolite crystallisation will extract Fe and produce a more differentiated melt composition. However, could this not be a local effect on melt composition, potentially forming boundary layers around crystals, as opposed to the residual melt? The experiments crystallise only 1 vol% nanolites, so the effect on melt composition may be localised and restricted to the environment surrounding the nanocrystals. How representative is the differentiated melt composition presented in Table 1 of the residual melt following the crystallization of nanolites? If the chemical effect is localised, how would this produce an order of magnitude increase in the bulk magma viscosity?

Reply: The possibility of a localised, more differentiated melt can be ruled out considering the DSC analyses. A localised effect such as a boundary layer would produce at least two different melt compositions in the samples, and hence glass compositions, that would be detected by the DSC analyses as a doubled or anomalously wider glass transition peak, considering that the amount of iron extracted will produce a marked difference between a potentially iron-poor boundary melt compared to a still iron-rich more distant melt. This is not observed in all the results of the DSC analyses after all the different stages of nanolites growth accounted in this work, but a clear homogeneous melt (glass transition) is observed in all the analyses instead (Fig. 2). Based on this, we believe that the compositions represented in Table 1 are good representatives of the melt compositions, and that there is no localised effect.

Line 205: Would the importance of melt iron extraction in increasing bulk viscosity change for more mafic magmas with higher iron content? The experiments in this study use a composition with only ~3 wt.% Fe₂O₃, whereas a more mafic composition could have over 10% total FeO. As these magmas are more Fe- and Ti-rich and prone to nanolite crystallisation, would the impact of the solid nanoparticles on the bulk viscosity still remain low? It is likely that these compositions would produce a higher nanolite content than the experiments. The results of Di Genova et al. (2020b) should be discussed here and the role of iron extraction and nanolite crystallisation for mafic compositions. The effect on more mafic compositions is

also referred to in the conclusion, it would be great to instead develop this point further here, before reaching the conclusions section, and discuss in greater detail the implications of the results for more Fe-rich, mafic compositions.

Reply: The higher iron contents of mafic magmas make them, indeed, more prone to oxide nanolites formation, forming a higher amount of nanolites that will be suspended in the fluid. It is expected that the influence of nanoparticles in suspension increases, also by the apparent higher potential of nanoparticles to form aggregates in these magmas. We have now discussed these effects in the text, particularly in the previous paragraph, including the reference suggested by the reviewer, as well as in previous paragraphs. Additionally, we have added further observations (Morrison et al., 2019; Valdivia et al., 2023) that relate a viscosity increase in basaltic magma to the potential formation of oxide nanolites.

Line 242: The reference to non-aggregated particles here implies that the lower contribution of the particle suspension to bulk magma viscosity in this study could also be explained by the lack of aggregation for the nanolites. As aggregation has been shown to occur in natural nanolite-bearing samples, it would be best to clarify in which cases melt iron extraction becomes the dominant process causing the viscosity increase and the particle suspension a secondary process. Is this the case for particle suspensions where there is no aggregation?

Reply: This has already been answered in the reply to previous comments. We would like to mention that we have added a discussion paragraph on particle aggregation to the text.

Line 249: This sentence does not specifically refer to the process which would increase viscosity for more mafic, Fe-rich compositions more prone to nanolite crystallisation, and I would suggest clarifying if the authors mean there is a greater degree of iron extraction from the melt for these magmas and to contextualise this with reference to the particle content. This comparison could be introduced earlier in the discussion and developed further.

Reply: We have clarified this concluding sentence adding reasoning to the statement (underlined). It now reads: **“We suggest that oxide nanolites crystallisation can be a process that significantly increases melt viscosity and subsequently magma bulk viscosity in more mafic magmas due to their commonly higher iron contents compared to the ones of this study, which make such magmas more prone to oxide nanolites formation, potentially forming a higher nanolite content and extracting a higher amount of iron from the melt phase”**.

Lines 306: Could the results from the textural analysis be presented in a table, possibly in the supplementary information?

Reply: We have added the results of the textural analyses to the Supplementary Information as suggested by the reviewer.

Line 311-312: Further detail is required for the EPMA analysis here. What standards were used for the analysis and what were the errors for each element analysed? Were the possible effects of glass devolatilisation evaluated and taken into account when acquiring the data? Please add these details to the supplementary information.

Reply: We have now added the required information regarding the EPMA analyses. We have considered the potential devolatilization of both alkali elements and water in the samples and minimize it by performing the analyses at proper conditions for such purposes. We have added all the details of these conditions into the manuscript in the methods section.

References

Di Genova, D., Zandona, A. and Deubener, J. (2020b). Unravelling the effect of nano-heterogeneity on the viscosity of silicate melts: Implications for glass manufacturing and volcanic eruptions. *Journal of Non-Crystalline Solids*, 545, 120248.

We wish to thank the reviewer for comments that helped us to improve the quality of the manuscript, as well as to discuss about the potential effects of particle aggregation on the viscosity of natural magmas.

Reviewer #2:

Review of “Oxide nanolite-induced melt iron extraction causes viscosity jumps and enhanced explosivity in silicic magma” by Cáceres et al. for Nature Communications

This manuscript presents the results of crystallization experiments and viscosity measurements on a series of melts containing iron oxide nano-crystals. The goal is to determine the effect of ~1 volume percent nanocrystals on magma viscosity, and to determine if this effect is primarily due to the physical effect of crystals or the chemical effect on changing the residual melt composition.

The data are technically sound, obtained with appropriate techniques such as electron microscopy and magnetic hysteresis, differential scanning calorimetry, and viscometry. The data have been analysed and interpreted carefully, and methods and results are both generally presented in sufficient detail, with the exception that uncertainties need to be calculated and plotted for the extrapolated melt viscosity in Figure 4C, on which much of the conclusions depend. Additional discussion of the temperature-time history of samples during the micropenetration viscosity experiments would be helpful in determining the possibility of nanolite formation during the measurements of the synthetic samples.

The conclusions generally appear to be valid, in that relatively small amounts of oxide crystallization can produce surprisingly high increases in viscosity. However, several claims are written in overly broad ways, for example “we show that the effect of particles (crystals) suspension is negligible in natural silicic magmas” is a vast overreach, and the idea of a maximum threshold value for the effect of nanocrystals is hardly justified based on one extrapolated datum with significant uncertainties.

Finally, the results are potentially significant and will be of interest to the community, because order of magnitude variations in viscosity do have important effects on magma dynamics. The role of nanoparticles has been a topic of intense interest for a few years now and this contribution sheds welcome additional light on how they affect the rheology of rhyolitic magmas, suggesting that in this case chemical effects are more important than physical effects.

Four specific topics that should be considered by the authors are listed below:

[1] Crystallization during viscosity and DSC measurements

Lines 135-142 “After viscosity analyses by micro-penetration method, the samples show crystallinities of ~1.12 vol.%, ten times higher than those observed in the samples right after the initial cooling-controlled experiments (~0.11 vol.%)” while “the rapid-quenched sample that shows an increase from none (nanolite-free) to a volume fraction ~0.61 vol%”.

This implies that the volume fraction of nanolites grew in all samples during the micro-penetration viscosity experiments. From the methods section, heating from 25 to 875°C took

34 minutes, followed by an isothermal hold of 15 minutes before measurement. How long did the viscosity measurements take, and was there any indication of an increase in viscosity during the measurement time? Furthermore, why was only one viscosity data point collected per sample – why not measure at 850, 875 and 900°C, which would better constrain the temperature-dependence of viscosity?

Reply: All micro-penetration analyses were performed at the same temperature, representative of magmatic temperatures for silicic magma: 875°C. The time of the isothermal hold before analyses was made in order allow relaxation of the melt, as well as to equilibrate any possible crystallization occurring at such temperature and having viscosity measured in a sample with a given unchanging crystal content. Crystallisation, indeed, occurred in the samples during the thermal history of the analyses as noticed by the reviewer and as shown in both Raman and magnetic analyses in Figure 1, however, this was not observed to occur during the viscosity analyses. This is supported when comparing the difference between the significant crystallization event occurred during the first heating in DSC analyses and the crystallisation event after micro-penetration, which can be seen in the magnetic analyses of Figure 1: the increase in crystal content after MP measurements compared to after DSC analyses is minor and it should have occurred latest during the isothermal phase of the micro-penetration analyses, since the 15 minutes isothermal time also represents a fully relaxed melt. This was added to the manuscript in order to clarify the reader the conditions of analyses and state of the samples during and after measurements. The duration of MP analyses was indeed 15 min after the first 15 min isothermal and they did not show any change in viscosity during the measurements.

On the other hand, only one viscosity data point was measured at the magmatic temperature of 875 °C for each sample in order to avoid further crystallisation when changing the measurement temperature, for instance to lower temperatures such as 850, 825 or 800 °C. Fixing the analyses at one given temperature allowed us to determine the exact nanolite content that crystallised at that given temperature and then establishing the relationship between viscosity and nanolite content. In other words, changing to lower temperatures naturally would have induced further crystallisation of the samples, not being possible to determine the crystal content at each viscosity measured.

Lines 150-151 “during fast heating both nucleation (for rq sample) and growth (for cc samples) occur in timescales as low as ≥ 1 s.” But this is not true for the rq sample, which only shows a slight hint of crystal nucleation on its fifth time being heated above T_g (Fig 2F). So slow cooling stage is very effective for crystal nucleation, and rapid heating is not. I agree with the authors that rapid heating is more effective for crystal growth (lines 151-153; Fig 3D).

Reply: We thank the reviewer for pointing the imprecision in the statement. In fact, slow cooling is effective for crystal nucleation and not for crystal growth, while rapid heating is effective for crystal growth and not for crystal nucleation. We have now re-written this sentence and it now reads: “**Here we observe that fast heating is effective for growth of nucleated crystals (for cc samples), occurring in timescales as low as $\geq 10^0$ s, but ineffective for crystal nucleation, evidenced by the originally crystal-free melt (rq sample) that nucleated slightly only during the fifth heating cycle of DSC analyses (Fig. 2F). On the other**

hand, slow cooling shows to be effective for nucleation of crystals, but not for their growth in silicic magma”.

Lines 187-192. Were the synthesized glasses checked for nanolite crystallization after the MP measurements? If the original natural rhyolite crystallized 0.6 vol% nanolites during the MP experiment, the synthetic rhyolites would likely also crystallize, but in different amounts based on their iron contents. That would affect the interpretation somewhat, although I doubt the overall conclusions would change.

Reply: Since the synthesised glasses were obtained from melts previously quenched from 1600 °C and that contain similar or less iron contents than the natural sample that were checked with Raman spectroscopy and Magnetic Hysteresis analyses, we can state that they are originally crystal-free previous micro-penetration analyses. Furthermore, two cylinders were obtained from the synthesised glasses, from which one was used for DSC determination of the glass transition and the second one was used for micro-penetration analyses performed at the T_g interval previously measured from the same glass material. Based on all this, and the fact that the natural composition showed no nucleation of oxide nanolites until several heating and cooling intervals at DSC (Fig. 2F) and several degrees hotter than the T_g interval, we are confident that the micro-penetration analyses performed within the T_g interval remain all crystal-free after viscosity measurements. This procedure, indeed, was followed in order to avoid oxide nanolites formation during the viscosity analyses and preserve the melts crystal-free. We recognised that this was previously not stated in the manuscript and we have now added it into the Methods section.

[2] Viscosity models and measurements

Perhaps the single most important datum in the paper is the blue circle at 10⁸ Pas on Figure 4C. Please check it is plotted in the right place.

Inspection of Fig 4A suggests the blue dot should be at 10^{8.2} Pas (and 10^{8.15} is quoted on line 174), but it is plotted closer to 10^{8.0} Pas on Fig 4C. It should also be indicated that this point is extrapolated from data collected at a much lower temperature, using calorimetry data that was then converted to a viscosity value (the circles in Fig. 4A).

The uncertainties in this extrapolated viscosity point need to be stated explicitly, and preferably given to 2 sigma, because the magnitude of the effect of microlites (i.e. the vertical difference between the circle and the triangles on Fig 4C) has the same uncertainties. This blue circle should have error bars on Figure 4C, because they are certainly larger than symbol size. Please also provide error bars for the triangles, which represent direct measurements. An uncertainty of ±0.1 log unit is mentioned in the text on line 170 but it is not clear if this is the measurement uncertainty.

Reply: We thank the reviewer for pointing out this mistake in plotting the blue dot. This was, indeed, a mistake in the plotting of the data point, since the value of 8.15 log units quoted in the text, as well as the extrapolated line in Fig. 4A are both correct, yet in Fig. 4C it was slightly shifted to a lower viscosity value. We have now corrected this data point and we have additionally re-checked all the values plotted in the figures to be correct.

We have now added uncertainty values as error bars in the data points of the figures, this by reducing the original symbol size, since errors were initially equivalent to symbol sizes. The data points in the figures are now represented with their uncertainties associated explicitly stated, allowing a better interpretation of the results and extrapolation of such results in the figures. WE have added explicitly that in Fig. 4C the blue dot represents an extrapolated value, as requested by the reviewer. The uncertainty of these extrapolated value is considered as the uncertainty of the DSC-derived viscosity data points. These DSC-derived viscosity data points were calibrated to the micro-penetration viscosity values of the synthesised melts, that were used for adjusting the shift factor to the compositions used in this study. This is now stated explicitly in the manuscript text, as well as pointed out in the caption of Figure 4. We have additionally stated explicitly in the manuscript the associated uncertainties of all viscosity measurements (see Methods section).

Line 349 / Figure S1. On the topic of converting DSC data to viscosity values, the shift factor is quoted as 0.29 log units but it is clearly slightly different for each of the four melts, so it would be better to give both the mean value and the range. These uncertainties then propagate into the VFT fit, and to its extrapolated value at 875°C.

Reply: These given value of 0.29 represents the average difference between the calculated viscosity (shift factor included) from the DSC analyses according to Gottsmann et al. (2002) and the actual measured values of viscosity by micro-penetration for the iron-bearing synthesised melts. This was made to reduce and better constrain the big associated uncertainty in such method, that were previously big for silicic magmas. These difference values range indeed between 0.26-0.32 for the iron-bearing melts and they averaged 0.29, with a standard deviation of 0.02. We have now added the range to the text as suggested by the reviewer, as well as the average and standard deviation, and we have explained how we have considered the associated uncertainties from the DSC-derived viscosities and their extrapolated values.

[3] Over generalization of findings.

Lines 176-177 “Here we show that the effect of particles (crystals) suspension is negligible in natural silicic magmas”. There needs to be a qualifying statement here – the highest crystal contents in this study are 1.1 volume % and such a low volume fraction would be expected to have a negligible effect on bulk viscosity (using any of the models reviewed by Mader et al. 2013 JVGR, for example). To then say that the effect of crystals is negligible in natural silicic

magmas (which can include crystal-rich dacites and rhyolites with over 50 volume % crystals) is a vast overreach. Something like “In our experiments, the effect of ~1 volume % particles (crystals) suspension is negligible, as expected based on decades of experimental work on suspension rheology”, with appropriate citations, would be better.

Reply: We acknowledge that this sentence may have been easy to misinterpret considering the strong effect that particle suspension exerts in crystal-rich silicic magma as stated by the reviewer and several studies. We have re-phrased this sentence in order to avoid an overgeneralisation of our findings, stating particularly the effect of nanoparticles suspension when nanocrystals form. This sentence now reads: **“Here we show that the effect of nanoparticles (nanocrystals) suspension may be negligible in natural silicic magmas, where low amounts of nanocrystals tend to form, such as those in this study (~1.12 vol%), and that the increase of magma bulk viscosity when oxide nanolites form is driven mainly by the increase of melt viscosity due to chemical changes of the melt phase, particularly iron extraction”**.

Lines 206-206 “we show that the initially rapid increase of magma viscosity with an increasing nanolite content diminishes considerably at higher nanolite contents, tending to a maximum threshold value”. This conclusion depends strongly on the uncertainties attached to the points in Fig 4C, but it also seems unlikely to be correct. In general higher crystal contents lead to higher viscosities, for both chemical and physical reasons. There is no reason why there would be a maximum threshold for the effect of crystals, whether large or small, on magma viscosity.

Reply: We agree with the reviewer that an increase in viscosity driven by the effect of crystals (particle suspension), this tends to increase without an apparent maximum. However, here we can argue that statement in two directions. On the one hand, we show that for natural systems the increase in viscosity is not only driven by a constant, infinite load of crystals, but it is dependent on the potential of such melt composition to form crystals, where the amount of possible crystals formed depends on the chemistry of the melt and the consumption of available elements. When consumed, the chemical components extracted from the melt phase will control the ability of a magma to form more crystals. On the other hand, indeed the increase in viscosity driven by a load of suspended crystal can be limited by the maximum packing at which crystals can organize and distribute in the suspension, yet this effect is not playing a role in our studied system given the very low crystal contents involved. So, the referred maximum threshold is here meant for natural magmas, where melt changes during crystallisation, and given by the ability of the melt chemistry to form crystals. We believe this is stated in the manuscript in such paragraph, that reads: **“Consequently in consequence, it can be inferred that the increase in viscosity due to oxide nanolites formation is limited by the total iron content that is extracted and initially available in the melt from where nanolites crystallise. Here we show that the initially rapid increase of magma viscosity with an increasing nanolite content diminishes considerably at higher nanolite contents, tending to a maximum threshold value (Fig. 4C). We suggest that this threshold is different for each magma composition and dependent on the amount of total iron that can be extracted”**.

[4] Additional support from previous results

Lines 249-251 several experimental rheology studies have found surprisingly large viscosity increases in basaltic magma viscosity at relative low crystal fractions (observed at micro but not nano scale).

See discussion in Morrison et al. 2019 Icarus, for example. Perhaps nanocrystals can explain these results, because the effect appears to be of similar magnitude (~1 order of magnitude viscosity increase). This may belong around line 210-217 instead, and would strengthen the final sentence of the manuscript's conclusion.

Reply: We thank the reviewer for providing new supporting information for the evidences and results of this study. We have included the reference suggested by the reviewer in the manuscript text, which, as said by the reviewer, strengthen the conclusions of this study.

Minor comments:

Line 179 it is not immediately clear what the 1:6 ratio is, especially as it is written as “crystals suspension : melt”. I am not sure if the authors intend to say “the increase of magma bulk viscosity is driven mainly by the increase of melt viscosity due to chemical changes of the melt phase, particularly iron extraction, accounting for 6-fold increase in the viscosity of the suspension relative to the uncrystallized melt.” Or whether they meant “the increase of magma bulk viscosity is driven mainly by the increase of melt viscosity due to chemical changes of the melt phase, whose effect is 6 times greater than the physical effect of the rigid crystals.”

Reply: In this sentence, we mean to express the second. Particularly, that the increase of magma bulk viscosity is driven mainly by the increase of melt viscosity due to chemical changes of the melt phase, whose effect is 6 times greater than the physical effect of the rigid crystals. We have re-written this sentence since it was not clear in its meaning. It now reads: **“Here we show that the effect of nanoparticles (nanocrystals) suspension may be negligible in natural silicic magmas, where low amounts of nanocrystals tend to form, such as those in this study (~1.12 vol%), and that the increase of magma bulk viscosity when oxide nanolites form is driven mainly by the increase of melt viscosity due to chemical changes of the melt phase, particularly iron extraction. In this study, these effects account for a ratio of ~1:6 between the physical effect of suspended crystals and the increase of melt viscosity, respectively.”**

Line 277 Change “form” to “from”

Reply: We have corrected this misspelling.

Line 585 Correct spelling of “triangles”

Reply: We have corrected the misspelling of the word.

Line 595 I suggest removing the dotted line. The reader does not need a line to guide the eye, there are three large symbols to do that already. There are other lines that could be drawn between these symbols; in this case, to choose one does not guide so much as impose an interpretation, especially because the uncertainties are not shown for the first symbol.

Reply: We thank the reviewer for this suggestion. Re-considering the drawing of the dotted line as a guide, we agree with the reviewer and we have now removed it in order to avoid any misguidance in the reading of the path followed by the increase in viscosity, behaviour shown by the results that still remains clear without such line.

Lines 610-613 Scarani et al. 2022 Comms Earth Env also suggest strong rheological effects of nanolites on basaltic magma viscosity. This is appropriately discussed in the main text, around line 210-217, but it does not also belong in the caption to figure 5.

Reply: We have removed the statement from the caption of Figure 5, following the suggestion of the reviewer.

We wish to thank the reviewer for insightful comments that helped to improve the quality of the manuscript. Particularly, allowing us to provide statistical uncertainties that prove the main statements and conclusions of this work with high quality quantitative data, making of this set of experiments a robust piece of work.

REVIEWERS' COMMENTS

Reviewer #1 (Remarks to the Author):

I have reviewed the revised version of the manuscript and I think the authors have done a great job in addressing the points of the reviewers and clarifying and developing parts of the discussion in further detail. I am happy for the article to be accepted in its current state.

Reviewer #2 (Remarks to the Author):

The authors have responded to both sets of comments made in the first round of peer review, and while I still do not agree with every nuance of their interpretation, the clarity of presentation and robustness of interpretation is much improved.

The legend for Figure 4B is likely to be misinterpreted, and should probably be changed. The "100% Total iron" line refers to the synthetic sample with 3.19 wt% Fe₂O₃, which would be a much better label for it and so on for the others - 2.35, 1.68, 0.83 and 0.08 wt% according to Table 1)

I recommend publication, especially if the legend in Figure 4B is changed.

In what follows we list the reviewers' comments in grey colour and our replies in blue colour.

All changes made to the manuscript file, based on the reviewers' comments and additional changes that we believe improved the quality of the manuscript, appear with track changes (underlined in red colour) within the corresponding manuscript file.

Reviewer #1:

I have reviewed the revised version of the manuscript and I think the authors have done a great job in addressing the points of the reviewers and clarifying and developing parts of the discussion in further detail. I am happy for the article to be accepted in its current state.

Reply: We thank the reviewer for recognising the improvements in our last version of the manuscript.

Reviewer #2:

The authors have responded to both sets of comments made in the first round of peer review, and while I still do not agree with every nuance of their interpretation, the clarity of presentation and robustness of interpretation is much improved.

The legend for Figure 4B is likely to be misinterpreted, and should probably be changed. The "100% Total iron" line refers to the synthetic sample with 3.19 wt% Fe₂O₃, which would be a much better label for it and so on for the others - 2.35, 1.68, 0.83 and 0.08 wt% according to Table 1)

I recommend publication, especially if the legend in Figure 4B is changed.

Reply: We would like to first thank the reviewer for recognising the improvements in our last version of the manuscript.

We have changed the legend in Figure 4B following the reviewer's suggestion and it is now stated according to the total iron content. We have used for this the normalised values of total iron of Table 1. We have also explained these values of the legend in the figure caption.